# Heterogeneous Dendrimer-Based Catalysts

**DOI:** 10.3390/polym14050981

**Published:** 2022-02-28

**Authors:** Eduard Karakhanov, Anton Maximov, Anna Zolotukhina

**Affiliations:** 1Department of Petroleum Chemistry and Organic Catalysis, Moscow State University, 119991 Moscow, Russia; kar@petrol.chem.msu.ru; 2Institute of Petrochemical Synthesis RAS, 119991 Moscow, Russia; max@ips.ac.ru

**Keywords:** dendrimers, nanocatalysis, heterogeneous catalysis, supramolecular chemistry, hydrogenation, cross-coupling

## Abstract

The present review compiles the advances in the dendritic catalysis within the last two decades, in particular concerning heterogeneous dendrimer-based catalysts and their and application in various processes, such as hydrogenation, oxidation, cross-coupling reactions, etc. There are considered three main approaches to the synthesis of immobilized heterogeneous dendrimer-based catalysts: (1) impregnation/adsorption on silica or carbon carriers; (2) dendrimer covalent grafting to various supports (silica, polystyrene, carbon nanotubes, porous aromatic frameworks, etc.), which may be performed in a divergent (as a gradual dendron growth on the support) or convergent way (as a grafting of whole dendrimer to the support); and (3) dendrimer cross-linking, using transition metal ions (resulting in coordination polymer networks) or bifunctional organic linkers, whose size, polarity, and rigidity define the properties of the resulted material. Additionally, magnetically separable dendritic catalysts, which can be synthesized using the three above-mentioned approaches, are also considered. Dendritic catalysts, synthesized in such ways, can be stored as powders and be easily separated from the reaction medium by filtration/centrifugation as traditional heterogeneous catalysts, maintaining efficiency as for homogeneous dendritic catalysts.

## 1. Introduction

Various catalytic processes, such as hydrogenation, oxidation, polymerization, etc., are of great importance in the modern petrochemistry and pharmaceutical industry [1,2,3,4,5,6]. The last tendencies, concerning the advanced science and technology in catalysis, consist in the transition to nanoscale and nanoreactor supramolecular systems, providing higher reaction rates and selectivity due to the metal nanoparticle high specific surface area, size, geometry, and electron effects as well as of specific microenvironment metal complexes or nanoparticles [7,8,9,10,11,12,13,14,15,16,17,18,19,20,21,22,23,24,25]. To prevent metal nanoparticle sintering and agglomeration, and, as a consequence, catalyst deactivation during the process, various organic ligands, such as ionic liquids [26,27,28,29,30,31,32,33,34,35], simple amines, phosphines, and thiols [16,26,27,28,29,30,31,32], cyclodextrins [24,26,35,36,37,38,39,40,41,42,43], calixarenes [26,44,45,46], and polymers were suggested [24,26,27,28,29,30,31,32,35,37,38,39,43,47,48,49,50,51,52,53,54,55,56]. Worthy to distinguish among the latter are dendrimers, three-dimensional globular macromolecules with a branched regular structure [10,57,58,59,60,61,62,63,64].

Dendrimers were independently synthesized for the first time by Vögtle [65,66], Denkewalter [67], Tomalia [68], Fréchet [69] and Newkome [70] in the 1980s. Since that time, due to their unique structure and relative facility for modification, dendrimers have found a wide application in nanomedicine [25,61,62,63,71,72,73,74,75,76,77,78], light and electron-sensing devices [25,63,71,72,76,79,80,81,82,83], material science [10,24,25,61,62,63,64,72,76,77,78,79,80,82,83,84,85,86], and, especially, in nanocatalysis [10,24,25,26,28,60,61,62,63,64,72,79,81,87,88,89,90,91,92,93,94,95]. Hence there were developed materials, based on poly(amido amine) (PAMAM) (Tomalia type) [68,96], poly(propylene imine) (Vögtle type) [66,81,97], poly(ether imine) (PETIM) [98,99,100,101,102], poly(ester amide) (Newkome type) [70,103], poly(benzyl/aryl ether) (Fréchet type) [68,104], poly(aliphatic ester) (Newkome type) [105], poly(carbosilane) [106], triazol or allyl-ended carbosilane arene-cored (Astruc type) [107,108], phenylazomethine (DPA) (Yamamoto type) [80,109], arylphosphite phosphazene cored (Caminade/Majoral-type) dendrimers, etc. (Figure 1) [110,111,112,113].

In the dendrimer structure, there is a core, branches, and periphery (terminal, end, or surface) groups (Figure 2) [10,26,63,64,68,71,79,80,85]. Herein PAMAM dendrimers of higher generations (G) are comparable in the size (up to 10 nm) with protein molecules [75,84,87,114].

The main advantages for dendrimer application in catalysis are: (1) the possibility to sorb a well-defined quantity of metal through the coordination by donor node and end groups [10,26,60,62,70,81,82,83,91,93,115]; (2) the control of substrate selectivity via choice of the dendrimer appropriate generation [25,26,60,92,93,116,117,118] and/or by terminal group modification, specifying the dendrimer solubility in the reaction medium and substrate affinity [25,60,62,73,80,90,92,93,116,117,118]; (3) the recyclability of dendrimer-based catalysts via fractional precipitation from solution or separation to another phase by the addition of appropriate, so-called “bad” solvents [60,89,93,108,115,119,120,121,122,123,124,125,126,127,128,129,130], or cooling the reaction mixture below upper critical solution temperature (UCST) or heating above low critical solution temperature (LCST) [63,71,77,89,90,94,115,131,132,133,134,135]. The active metallocenters in dendrimers can be located in a core or at the focal point (Figure 3a) [10,60,62,64,79,84,89,90,92,93,113,119,120,121,129,131,136,137,138], at nodes (Figure 3b) [10,60,64,80,93,113,132,136,139,140], end functional groups (Figure 3c) [60,63,79,89,90,92,93,94,101,102,108,113,136,140,141,142,143,144,145,146,147,148,149,150], or in a cavity between dendrimer branches (Figure 3d) [10,60,62,64,90,93,94,115,116,117,118,125,126,136,139,150].

In the end of 1990s and 2000s, Crooks and co-workers [10,25,26,28,60,62,64,91,92,115,117,151,152,153,154], on the one hand, and Kaneda and Mizugaki [10,26,28,62,64,91,92,116,155,156,157], on the other hand, suggested an approach for the synthesis of Cu, Pt, Pd, Rh, Ag, and Au nanoparticles (NP) of 1–3 nm in diameter, encapsulated inside the cavities of PAMAM or PPI dendrimers. This approach includes two steps (Figure 1): (1) metal ion complexation with the dendrimer functional groups (both nodes and periphery); and (2) reduction by NaBH_4_ or KBH_4_ (sodium or potassium borohydride), resulting in nanoparticle or cluster formation. It can be applied for synthesis of both monometallic and bi- or polymetallic dendrimer-encapsulated nanoparticles [10,25,62,64,91,92,95,115,158].

In the last case, one may distinguish three different sub-approaches (Figure 2). The first one is co-complexation of two or more different metal ions simultaneously, resulting in alloy or core-shell nanoparticles depending on the reduction rate of each metal (Figure 2) [10,62,64,88,91,92,95,115,131,154,158,159,160,161,162,163,164,165,166]. The second one is the subsequent complexation, including two repeated procedures of complexation followed by reduction for each metal and, as a rule, resulting in core-shell nanoparticles (Figure 2) [10,62,64,88,91,95,115,154,158,167,168]. The third one is the so-called galvanic displacement (Figure 2), suggesting the use of oxidizable metal (e.g., Cu^0^) as a reducing agent for more noble metals with higher oxidation-reduction potential (e.g., Pt, Au), resulting in core/shell or alloy-type bimetallic nanoparticles [88,91,95,115,158,169,170].

The driving force, providing the metal ion adsorption, uniform distribution, and well dispersion in the interior of the dendrimer molecule, further resulting in the formation of metal nanoparticles and clusters with narrow size distribution, was here for the coordination and complex formation between metal ions and dendrimer amino groups [60,88,115,116,117,151,155,156]. Moreover, dendrimer-encapsulated nanoparticles and clusters may be formed from dendritic metal complexes in situ under hydrogenation conditions [140,148,171] or via partial reduction by NH_2_ end groups of PPI or PAMAM dendrimers [128,148,171,172,173,174,175,176,177]. As a rule, due to the steric hindrances, the mean particle size decreased with the increase in dendrimer generation [91,116,155].

The similar approach, including metal ion complexation followed by reduction with NaBH_4_, was applied by Yamamoto’s group for the synthesis of ultra-small mono-, bi-, and polymetallic nanoparticles and clusters (of 0.5–1.5 nm in diameter), encapsulated into the cavities of poly(phenylazomethine) dendrimers [10,64,162,178,179,180,181,182,183,184]. It is worth noting that, in contrast to globular PPI, PAMAM, and other related poly(alkyl amine) dendrimers, the complexation of rigid poly(phenylazomethine) dendrimers (DPA) with metal ions proceeds radially, layer by layer, from core to periphery (Figure 3) [10,64,80,109,166,178,179,180,181,182,183,184,185]. This is due to the rigid, conjugated, electron transfer structure of poly(phenylazomethine) dendrimers, resulting in higher electron density and, as a consequence, in higher proton association constants for core imine groups, as compared with periphery [10,80,109,162,183,184,185].

With regard to bi- and polymetallic complexes, it implies the preferential complexation of inner imine groups by metal ions with stronger association constants, whereas less affine metal ions coordinate outer imine sites [10,109,166,183,184,185]. Reduction of bi- and polymetallic DPA complexes with sodium borohydride afforded to formation of alloy-type nanoparticles and clusters [166,183,184,185].

Astruc et al. developed hybrid aryl-cored poly(carbosilane/triazol/aryl ether) dendrimers, terminated with allyl, ferrocenyl, triazol, sodium sulfonate, or tri(ethylene glycol) moieties, and used them for the synthesis of Pd, Cu, and Au nanoparticles with sodium borohydride, MeOH, or sodium ascorbate as reducing agents [90,107,108,150,186,187,188,189,190,191]. Herein, depending on the dendrimer size and generation, as well as on the metal to dendrimer ratio, small dendrimer-encapsulated nanoparticles (DENs) of 1–2 nm can be obtained for Pd with the narrower size distribution or dendrimer-stabilized nanoparticles (DSNs) with the sizes of 1–4 nm for Pd, 3–12 nm for Au, and 4–20 nm for Ag (Figure 4) [10,25,62,64,90,150,186,187,188,191]. The similar is also true for PPI, PAMAM, and other related poly(alkyl amine) dendrimers [60,115,116,156,168,192,193].

Dendrimer-stabilized core nanoparticles (or nanoparticle-cored dendrimers, NCD) were mostly synthesized in the presence of Fréchet type poly(aryl ether) or poly(aryl ester) dendrons with the donor phosphine, thiol, hydroxyl, or amino groups at the focal point, resulting in so-called monolayer protected cores (MPC) (Figure 5) [10,62,84,89,119,120,121,129,137,194,195,196,197,198,199,200]. The synthesis was usually carried out in the organic solvent (toluene or THF) as a metal source and reducing agent, respectively; the combinations of easily decomposed organic metal salt (e.g., Pd(OAc)_2_) or complex (e.g., [Ru(Cumene)Cl_2_]_2_ or [Ru(COD)(COT)]) and molecular hydrogen [119,120,129], or chloride complex salt or acid (e.g., K_2_PdCl_4_, HAuCl_4_ or H_2_PtCl_6_), often additionally stabilized by amphiphilic ionic liquid (e.g., [N(C_8_H_17_)_4_]Br), and sodium borohydride [121,137,194,195,196,197,198,199,200], were used. Another approach implied the modification of nanoparticles, already stabilized by bifunctional ligands (e.g., Au NPs, stabilized by 11-mercaptoundecanoic acid), with hydroxyl poly(aryl ether) dendrons of various generations (Figure 4), hence providing the nanoparticle uniform and narrow in size distribution [201].

Herein the mean particle size was strongly dependent on the nature of the donor group in the dendron focal point, reaching 1–6 nm for Au and Pd nanoparticles, coated with more affine thiol or phosphine dendrons [121,129,194,195,196,197,198], thus revealing the wider particle size distribution in comparison with PAMAM or PPI dendrimer-encapsulated nanoparticles [91,116,151,156], and 2–25 nm for Au-cored dendrimers with hydroxyl stabilizer at the focal point [199], and are thus comparable with monometallic Pd and Au nanoparticles, stabilized by PAMAM, PPI [168,192], or poly(carbosilane/triazol/aryl ether) dendrimers [188,191]. Pd nanoparticle core size decreased with the increase in the dendron generation [129], whereas Au and Ag nanoparticle core size increased [195,196,197,198,199,202], with the distribution, depending not only on the dendron generation but also on the reduction conditions [199].

Among the other nanoparticle-cored dendrimers, the synthesis of dendrimer core monometallic Pt and bimetallic core/shell Pt[Au] nanoparticles was also reported, stabilized by poly(aryl ether) dendrons with tris(acetic acid) ammonium moieties at focal points (Figure 6) [62,200,203]; of Au nanoparticles, stabilized by thiol poly(*L*-lysine) dendrons (Figure 7) [83,204]; of core/shell CdSe/ZnS nanoparticles, stabilized by thiol-focal allyl-terminated poly(ethylene imine)/poly(amido amine) dendrons, which further were linked together by metathesis reaction (Figure 8) [84,205,206]; of monometallic Pd and bimetallic PtPd alloyed nanoparticles, stabilized by PAMAM dendronized 4-(aminomethyl pyridine), etc. [131,207]. In the last case, both organic metal sources ([PdCl(C_3_H_5_)]_2_ and PtCl_2_(PhCN)_2_ in CH_2_Cl_2_) and reducing agent (LiB(C_2_H_5_)_3_H) were used, resulting in mono- or bimetallic nanoparticles with strongly reproducible mean sizes for the same dendron generation and periphery and decreasing with the increase in the dendron generation and the length of terminal moiety (Figure 5) [131,207].

Thus, synthesized metal complexes and nanoparticle dendrimer-based catalysts, both monometallic and bi/polymetallic, were successfully applied in various processes, such as hydrogenation of alkenes, phenols, aromatic and heterocyclic compounds, semihydrogenation of alkynes, dienes and polyenes, hydrogenation and hydroamination of carbonyl compounds, nitroarene reduction, oxygen reduction, hydroformylation, alkyne-azide cycloaddition, allylic addition, asymmetric synthesis and catalysis, Wacker oxidation, C–H, O–H, and sulfide oxidation and epoxidation, olefin oligomerization and polymerization, metathesis, oxidative coupling, and various cross-coupling reactions since the second half of the 1990s until the end of the 2010s (see Table 1) [10,24,25,26,60,62,64,72,79,84,88,89,90,91,92,93,94,113,114,115,136,145]. Herein reaction turnover numbers (TON) reached 2,700,000, while turnover frequencies (TOF) exceeding 28,000 h^−1^ [186,188]. Moreover, bi- and polymetallic catalysts, both metal complex [109] and nanoparticle [131,159,160,163,164,166,170,183,203], exhibited the enhanced efficiency due to the metal synergism and changed electron properties and geometry of nanoparticles, clusters, and complexes [10,62,109,115,164,183,203,208,209,210,211,212,213].

Depending simultaneously on the reaction type, dendrimer nature, and catalytic centers location, as well as on the overall weight of the stabilization or steric factors from the dendrimer ligand, the so called positive dendritic effect can be observed, appearing in the enhanced catalyst performance with the increase in the dendrimer generation [79,80,81,82,83,84,85,86,87,88,89,90,92,94,113,115,120,121,129,131,132,136,141,145,146,149,190,193,195,207,215,216,218,221,227,242,247,248,252,255,258,261,262,263,271,272,273,274,279,280,281,282,283] or the negative dendritic effect, respectively, and the drop in the catalytic activity with the increase in the dendrimer generation or substrate size [60,90,91,94,101,116,117,122,124,126,130,132,136,142,143,145,147,148,150,187,193,200,214,227,234,271,281,282,283,284].

Moreover, the dendrimer nature was found to have an essential influence on the selectivity in the hydrogenation of alkynes and dienes in the presence of Pd-containing catalysts. When Pd nanoparticle-cored Fréchet type poly(aryl ether) dendrimers were used, both covalently attached to the Pd surface [137] or stabilizing via phosphine focal point [129], C≡C triple bonds or conjugated C=C double bonds underwent exhaustive hydrogenation. Vice versa, alkynes and dienes could be selectively converted to corresponding monoenes in the presence of Pd catalysts (both mono- and bimetallic), based on triazol [150], PAMAM [131,148,159,160,207,241] and, especially, on PPI dendrimers [116,128,140,156,171,193,241], with the total selectivity on alkenes increased with the increase in the dendrimer generation [116,148,193]. It was due to the poisoning effects of pyridyl moieties at the dendron focal point and/or dendrimer node donor tertiary amino group, occupying a part of adsorption sites and thus enriching the nanoparticle surface with electrons, facilitating the de-adsorption of alkene formed and interfering with its possible further re-adsorption [4,128,131,171,177,193,207,285,286,287]. Herein, the product distribution in semi-hydrogenation of alkynes and, especially, dienes, was strongly affected by metal composition of the catalyst active phase; the dendrimer nature, influencing the metal surface electron properties, as well as substrate and intermediate adsorption features; and, finally, the substrate structure (including the accessibility of C=C double bonds) and electronic properties (the presence of electron donor (EDG) or electron withdrawing groups (EWG)) [128,131,150,171,193,207,288,289].

However, in spite of their remarkably high activity, selectivity, and recyclability [121,130,131,132,190,234,269], homogeneous dendrimer-based catalysts can be stored and used mostly as colloidal solutions, undergoing gradual deactivation due to metal leaching and precipitation from the dendrimer stabilizing micelles [118,126,164,186,187,278] or, otherwise, should be prepared in situ [130]. As a consequence, since the end of the 1990s, various approaches for heterogenization of dendrimer-based catalysts have been suggested. These heterogenized dendritic catalysts are supposed to exist as powders, easily precipitate, and isolate from the reaction medium using filtration or centrifugation—similar to conventional heterogeneous catalysts. In some cases, homogeneous micellar dendritic catalysts, initially obtained as solids, may be used as heterogeneous in the “bad” solvent medium [121]. Immobilization of dendritic catalysts, using various heterogeneous carriers, such as silica, polystyrene, etc., allows to significantly enhance their stability and recyclability.

Presently, three main approaches to the synthesis of immobilized heterogeneous dendrimer-based catalysts have been developed and are further considered in the present review.

The first is the adsorption of metal-containing dendrimers or dendrons on the surface of amorphous silica, carbon, or inside channels in channels of mesoporous materials via hydrophobic, electrostatic, or π–π interactions [86]. This approach is considered in Section 2.

The second approach supposes the covalent grafting of dendrimers or dendrons on polymers, amorphous silica, carbon nanotubes, or inner walls of mesoporous materials [86]. It is considered in Section 3.

The third approach, including the dendrimer cross-linking, using various transition metal ions, as well as bi- or trifunctional agents, polymers, etc., [241], is discussed in Section 4.

Additionally, magnetically separable dendritic catalysts [62], which may be designed using all three approaches mentioned above, are considered in Section 5.

## 2. Non-Covalent Immobilization of Dendritic Catalyst

One of the simplest methods for the immobilization of homogeneous dendritic catalysts, developed by Somorjai [10,64,290,291] and Yamamoto [10,64,178], implies the deposition of colloidal DEN solutions (mono- or bi/polymetallic) onto the various heterogeneous carriers, such as silica (both amorphous and mesoporous), zirconia, and carbon materials, by the wetness impregnation method (Figure 6). It is worth noting that the said approaches suppose the remaining of dendrimers inside the SBA-15 channels or on the carbon surface using electrostatic, hydrogen bonding, hydrophobic, or π–π interactions, hence preventing nanoparticle agglomeration sintering [178,292,293]—in contrast to previous works, where dendrimers were used just as templates for nanoparticle and pore formation and then removed under calcination [26,169,290,291,294,295,296,297,298,299,300,301]. Thus, well dispersed and tiny nanoparticles of 1.0–2.5 nm in diameter inside the SBA-15 channels [292,293] or subnanoclusters with the sizes of 0.5–1.5 nm, deposited onto mesoporous carbon [178,180,183,184], were obtained.

### 2.1. Impregnation/Adsorption on Silica-Based Carriers

Using the above-mentioned approach, Somorjai and co-workers synthesized highly active Pt and Rh nanocatalysts for ethylene (at room temperature and 10 atm. of H_2_) and pyrrole hydrogenation, including ring-opening reaction (at 60–150 °C and 50 atm. of H_2_) with turnover frequencies (TOF) up to 4.3 s^−1^ (~15,500 h^−1^) (Figure 9) [292,302]. Pd, Pt, and Rh nanoparticles, encapsulated into PAMAM dendrimers and supported on SBA-15, appeared highly active in the reversible dehydrogenation/hydrogenation of *N*-heterocycles, such as (tetrahydro)quinolines and (dihydro) indoles at 130/60 °C, resulting in the quantitative yields within 24 h [303]. Furthermore, PAMAM dendrimer-encapsulated Pt and Rh nanoparticles, supported on SBA-15, were successfully applied for liquid (at 20–80 °C) and gas-phase (at 200−225 °C) hydrogenative ring-opening and isomerization of methylcyclopentane (MCP) and alkyl/aryl cyclopropane (Figure 9) [304,305], whereas Pd nanoparticles, synthesized in the same manner, favored MCP dehydrogenation and ring enlargement to benzene at 250–275 °C, giving quantitative conversions and TOF values up to 335 h^−1^ [304], as well as effectively catalyzed electrophilic arylation of indole and one-pot tandem dehydrogenation/arylation of indoline at 60–130 °C (Figure 9) [303]. Herein, increases in the mean particle size and dendrimer generation lowered the reaction activation energy, thus favoring the enhanced catalyst performance [305].

Supported on SBA-15, Pt, Pd, and Au nanoparticles encapsulated in PAMAM dendrimers, in the presence of PhICl_2_ as an oxidative catalyst activator, exhibited high activity, just slightly inferior to that for homogeneous catalysts PtCl_2_ or AuCl_3_, in the electrophilic cycloaddition of *ortho*-alkynyl arenes, phenols, anilines, benzyl amines, or benzoate esters to C≡C triple bond, thus forming 5- or 6-membered heterocycles with the yields of 45–98%, depending on the substrate structure (20–100 °C, 15 h, toluene, Ar) (Figure 9) [306,307,308,309]. It should be noted that Pt@PAMAM-G4/SBA-15 appeared to be a much more stable catalyst in comparison with its analogue, stabilized by poly(vinyl pirrolidone) (PVP), and maintained its efficacy at reuse [306].

Moreover, the catalyst Au@PAMAM-G4/SBA-15 revealed the best performance and well stability in the cyclopropane synthesis via alkene-alkyne cycloaddition (20–70 °C, 12 h, toluene) (Figure 9) [310] and Hayashi-Ito aldol cycloaddition between aryl aldehyde and methyl isocyanoacetate (20 °C, PhICl_2_, 18 h, toluene), thus surpassing conventional homogeneous catalysts, based on Au carbene complexes (Figure 9) [311]. In the last case it was additionally established that larger pore size of the heterogeneous carrier favored the preferential *trans*-product formation, whereas smaller pore size constrained *cis*-product formation respectively, with a common decrease in the reaction rate for +*I* and −*M* substituents in the aryl aldehyde molecule, thus retarding electrophilic attack of the aldehyde group to enolate deprotonated forms of isocyanoacetate (Figure 10) [311].

It is worthy to note that heterogenized catalysts, based on PAMAM dendrimer-encapsulated nanoparticles loaded to mesoporous SBA-15 silica, can be easily adopted not only for batch, but also to continuous-flow conditions, exhibiting well time-on-stream and recycling properties [307,309,310].

The similar approach was suggested for the synthesis of PAMAM dendrimer-encapsulated Ir nanoparticles, immobilized on amorphous SiO_2_, and further applied for the hydrogenation of 2-nitrobenzaldehyde under the mild conditions (30 °C, 1 atm. of H_2_) [227]. Herein, the conversion of 98% was achieved within 1 h, and only the NO_2_ group in the substrate structure was subjected to hydrogenation. The increase in mean particle size from 0.9 to 1.5 nm resulted in the increased portion from 17% to 42% of anthranil, the product of 2-nitrobenzaldehyde semihydrogenation followed by a heterocyclization-dehydration reaction between the intermediately formed NHOH group and the C(=)OH group (Figure 7), whcih was attributed to more efficient dissociative adsorption of H_2_ on smaller Ir clusters [227].

Meijboom et al. proposed additional post-treatment of PAMAM dendrimer-encapsulated Ru nanoparticles, immobilized on amorphous silica, with ionic liquid solution; thus, the so-called solid catalysts with an ionic liquid layer (SCILL) were obtained [312,313]. The mean particle size was found as 2.1–2.4 nm, decreasing with the increase in the dendrimer generation from G4 to G6. The catalysts synthesized were studied in the hydrogenation of citral (90–130 °C, 10–30 atm. of H_2_, cyclohexane, 4 h) [312] and toluene (110 °C, 30 atm. of H_2_, cyclohexane, 4 h) [313]. The catalyst activity and selectivity were found to be strongly dependent on both the dendrimer generation and the ionic liquid nature, as well as on the reaction conditions used.

The main product of citral hydrogenation was citronellal (Figure 8) [312]. The latter was found out to undergo cyclization to isopulegol—especially at longer reaction times, elevated temperatures, and decreases in the dendrimer generation, which authors explained by thermal decomposition of PAMAM dendrimers, taking place at temperatures of 100–150 °C [115,312,314]. The best results (the selectivity on citronellal of 100% at citral conversion of 50% after 4 h) were achieved for the catalyst, based on the dendrimer of the 5th generation, and under the use of [BMIM][NTf2] ionic liquid as catalyst coating [312].

As for partial toluene hydrogenation, the selectivity on methylcyclohexenes at 110 °C and 30 atm. of H_2_ in the presence of [EMIM][NTf_2_] ionic liquid did not exceed 12%, 21%, and 16% at conversion of 9–13% for the catalysts, based on PAMAM dendrimers of the 4th, 5th, and 6th generations, respectively (Figure 11a) [313]. Further, for the catalysts, based on PAMAM dendrimers of the 4th, 5th, and 6th generations, the selectivities on methylcyclohexenes at conversions of 69–73% under the same conditions reached 8.5%, 7.5%, and 6% respectively (Figure 11b) [313]. Thus, one can observe, that in the beginning of the reaction, the selectivity on methylcyclohexenes initially increases from the G4 to G5 catalyst and then decreases from the G5 to G6 catalyst (Figure 11). The authors explained the results obtained in terms of mean particle size, increasing with the increase in the generation of dendrimer [313,315].

It should also be noted that C=C double bonds more easily undergo hydrogenation in the presence of basic promoters [312,316]. The basicity of the PPI or PAMAM dendrimer increased with its generation and the number of terminal primary and node tertiary amino groups increase [317,318,319]. Hence, the results, obtained after partial hydrogenation of toluene in the presence of Ru nanoparticles, encapsulated into PAMAM dendrimers and supported on silica [313], appeared to be noticeably inferior to those obtained for conventional heterogeneous catalysts, especially containing the bimetallic RuZn phase, supported on the acid solid oxide, such as ZrO_2_ or CeO_2_ [315,320,321,322,323].

### 2.2. Impregnation/Adsorption on Carbon-Based Carriers

Yamamoto et al. developed catalysts based on DPA-encapsulated subnanoclusters, supported on graphitized mesoporous carbon (GMC), Ketjenblack (KB), and zirconia, for effective hydrogenation and oxidation reactions (Figure 12) [10,64,178,179,180,183,245,246]. Herein, carbon supports, such as GMC or KB, provided effective catalyst adsorption and stability due to the π–π interactions between aromatic fragments of dendrimer and carrier, thus preventing metal nanocluster post-aggregation [178,179,245].

In particular, Pt_12_ subnanoclusters with the mean diameter of 0.9 nm appeared highly active in the hydrogenation of various alkenes, including unreactive substrates, such as 4-chlorostyrene and, especially, 4-trifluoromethylstyrene [178], as well as superior poison tolerance to amines in the reductive hydroamination of aldehydes [179]—in contrast to 2.2 nm Pt nanoparticles, synthesized in the absence of the dendrimer template. TOF values up to 5850 h^−1^ in the hydrogenation of alkenes and quantitative yields of hydroamination products were achieved in the presence of the heterogeneous dendrimer-based catalyst Pt_12_@DPA-G4/GMC, already under the ambient conditions within 5 h (25 °C, 1 atm. of H_2_, MeOH) [178,179].

In the presence of *tert*-butylhydroperoxide under ambient conditions (25 °C, 6 h, *n*-decane) [245] or molecular oxygen (160 °C, 10 atm. of O_2_, 5 h) [180] DPA-encapsulated Pt subnanoclusters, immobilized on graphitized mesoporous carbon or Ketjenblack, afforded the oxidation of secondary alcohols, such as 1-phenylethanol [245] or aromatic hydrocarbons, such as toluene, respectively [180]. In the last case, the mixture of benzyl alcohol, benzaldehyde, and benzoic acid was formed, and the maximum TOF values reached up to 3300 h^−1^ [180]. It was established that oxidation, in both cases, proceeded via radical-chain mechanism, depending on cluster/particle size and metal nature [180]. In particular, larger sized nanoclusters (>30 atoms) promoted the cleavage of the O–O bond of the adsorbed O_2_ or *tert*-butylhydroperoxide molecule, thus providing the charge transfer to the adsorbed substrate molecule [180,245]. At the same time, electron-deficient subnanoclusters (12–19 atoms) of less oxophilic metals, such as Pt, tended to one-end coordination of O–O bond, thus providing the formation of peroxide radicals on the surface with the subsequent one-electron transfer followed by C–H bond cleavage, hence initiating toluene radical oxidation processes similar to homogeneous or nanoheterogeneous molecular oxidation catalysts [180,324,325]. More oxophilic metals, such as Ru or Cu, and vice versa, were prone to form partial surface oxides, resulting in a drastic decrease in the catalyst activity, with TOF values not exceeding 100 h^−1^ [180].

Cu_n_O_x_ subnanoclusters, encapsulated in phenyl azamethyne dendrimers and immobilized onto ZrO_2_, appeared to be much more active catalysts under the same conditions [246]. The catalyst efficiency and TOF values, respectively, increased with the decrease in cluster size, reaching up to 720 h^−1^, which was due to the increased polarization of Cu–O bonds [246]. Nonetheless, the full conversion of toluene to benzoic acid was not completed, even after 128 h [246]. Introduction of additional metals, such as Pt and Au, in the cluster structure resulted in the essential increase in the catalyst activity in the oxidation of aromatic hydrocarbons (e.g., indane) by molecular oxygen, even under much milder conditions (90 °C, 1 atm. of O_2_, 6 h) [183]. The maximum TOF values, exceeding 600 h^−1^, were achieved for trimetallic CuPtAu clusters, that authors explained in terms of synergistic effects in alloy, arising from stabilization of Cu (0) and Cu (I) species by Pt, Ag, and Au, preventing the further oxidation of the former to Cu (II) [183,326,327]. It was established that oxidation of indane to 1-indanone proceeded mainly through the hydroperoxide stage, accompanied by Cu (0)/Cu (I) one-electron transitions on the interface between Cu, Au, and Pt [183].

### 2.3. Non-Covalent Immobilization on Modified/Coated Surfaces Using Hydrophobic Interactions or Hydrogen Bonding

Crooks and Stevenson suggested a method for the immobilization of PAMAM dendrimer-encapsulated monometallic Pd and Pt, or bimetallic PtPd nanoparticles on graphite or gold surfaces, using electrostatic interactions and hydrogen bonding [10,64,91,328,329]. For gold electrodes, α,ω-mercaptoundecanoic acid (MUA) was used as a non-covalent linker, and self-organization of near-surface charged monolayers took place (Figure 13) [26,91]. In the case of graphite electrodes, these interactions occurred between dendrimer terminal NH_2_ groups and -C(=O)OH groups, arising from defects on the graphite surface [328]. Hydrogen bonding provided the effective adsorption of PAMAM dendrimers, impregnated with Pt nanoparticles, on the surface of nitrogen-doped single-wall carbon nanotubes (SWCNT), synthesized by chemical vapor deposition (CVD) method (Figure 9) [329]. Thus, obtained systems proved their efficacy in the electrocatalytic oxygen reduction, that could effectively proceed only for small nanoparticles (<2 nm in diameter), and the maximum activity was found for bimetallic nanoparticles Pt_150_Pd_30_, encapsulated into PAMAM dendrimers of the 6th generation, adsorbed on the graphite electrode [64,328].

Replacement of 11-mercaptoundecanoic acid with 11-mercaptoundecyl-1-tri(ethylene glycol) and 2-(11-mercaptoundecyl-1-hexa(ethylene glycol)) acetic acid, modified by 6-(maleic acid imide)hexanoic acid hydrazide, followed by covalent attachment to PAMAM dendrimers impregnated with Au nanoparticles, resulted in a high stable sensor for selective insulin detection with a concentration limit as low as 0.5 pM (Figure 10) [62,330].

The more sophisticated approach was applied by Ghiaci’s group for the synthesis of bimetallic Mn/Co heterogeneous catalyst for *p*-xylene oxidation [324,331]. The negatively charged bentonite surface was treated here by cetyl pyridinium bromide; the hydrophobic tails of the latter retained the focal dodecyl moieties of PAMAM dendrons of the 1st generation, modified with salicylic aldehyde on the periphery (Figure 14) [331]. This bilayer system provided the location of Mn^2+^ and Co^2+^ ions, coordinated on the dendron salicylidene imine end groups, easily accessible for the *p*-xylene substrate. Hence, under the optimal conditions (KBr, AcOH, 190 °C, 17 atm. of O_2_, 3 h), a near to quantitative yield of terephthalic acid (97%) was provided. Nonetheless, in spite of the use of strongly coordinating chelating dendritic ligands, the catalyst synthesized appeared unable to effectively retain Mn^2+^ and Co^2+^ ions, resulting in the sharp downfall for the yield of terephthalic acid at recycling [331].

## 3. Covalent Immobilization on Heterogeneous Supports

Heterogeneous catalysts, based on PAMAM, PPI, poly(aryl ether), poly(ester amide), etc., dendrimers, and dendrons covalently attached to various preliminary functionalized insoluble supports, such as amorphous and mesoporous silica, polystyrene, carbon nanotubes, etc., have been developed since the end of 1990s [10,25,26,62,64,90,92,93]. Herein, dendrimer end group post-modification and metal deposition (both complex formation and nanoparticle encapsulation) is performed in the same way, as for homogeneous catalysts [26,146,147,148,149,171,193].

Covalent dendrimer/dendron grafting may be realized in both divergent (otherwise called “graft-from”) and convergent ways (otherwise called “graft-to”). The first one assumes the gradual, layer by layer, generation by generation dendron growth from the focal point, which is the functional group on the carrier surface (Figure 11) [10,25,26,62,63,71,82,83,86,92,113,114,147,324]. The convergent approach, in its turn, implies the covalent attachment of the ready-synthesized dendrimers or dendrons to the preliminary functionalized support, and can be performed through the dendron focal point or dendrimer end groups [10,25,62,63,74,92,102,114,171,193]. Both of these approaches are further considered in the present review.

### 3.1. Divergent Dendrimer Grafting

#### 3.1.1. Divergent Dendrimer Grafting on Silica Supports

Silica appears to be the conventional support for dendron grafting; as usual, it was preliminarily activated by (3-aminopropyl)triethoxysilane (APTES) treatment, followed by repeated divergent dendron growth (Figure 12) [92,93]. Thus, Chung and Rhee developed heterogeneous catalysts, based on PAMAM dendrons of different generations, attached to silica through 3-aminopropyl spacer, and applied them in the enantioselective addition of diethylzinc to benzaldehyde (0 °C, toluene, 48 h) [92]. The catalyst chirality was provided by the decoration of dendron end groups with (1*R*, 2*S*)-ephedrine (Figure 15). It was found out that at low dendron density, the catalyst activity, selectivity, and enantiomeric excess (ee) increased from G0.5 to G3.5 dendrons, revealing the positive dendritic effect, and then dropped back, when passing to the G4.5 dendron. Herein, SiO_2_-G3.5 catalyst in its performance was just slight inferior to homogeneous one and could be recycles at least three times without significant loss in effectiveness [92].

Kawi et al. synthesized epoxidation catalyst, based on Mn (II) complexes with salicylidene-terminated PAMAM dendrons of the 0–4th generations, grafted to silica (Figure 12) [252]. The yields of styrene oxide reached 20–75% after 1 h at 0 °C, using *meta*-chloroperbenxoic acid (*m*-CPBA) as an oxidant and *N*-methylmorpholine *N*-oxide as a co-oxidant in CH_2_Cl_2_ medium, rather than increasing with the increase in the dendron generation. Thus, the strong positive dendritic effect was observed. Nonetheless, the use of bulky substrates with an internal C=C double bond resulted in the noticeable decrease in the catalyst activity and reaction rate, which was typically for sterically crowded dendrimers of higher generations [60,91,117].

The similar approach for dendrimer immobilization was applied by Wu and Zhang to design hollow capsule material, impregnated with gold nanoparticles [26,333]. Herein, silica gel particles of ~260 nm in diameter, coated with grafted PAMAM dendrons of the 4th generation and already containing Au^0^ nanoparticles with the mean particle size of 2.3 nm, uniformly distributed over the material surface, were treated with poly(sodium 4-styrenesulfonate) and poly(allylamine hydrochloride), forming polyelectrolyte layers. Subsequent removal of silica core in the presence of HF resulted in hollow capsules of ~200 nm in diameter (Figure 13). Thus, the synthesized catalyst proved its effectiveness in the 4-nitrophenol reduction, giving the quantitative yields already within 30 min at room temperature and was reused 5 times without significant loss in activity [333].

Alper, with co-workers, suggested the decoration of grafted PAMAM dendron periphery by palladium or rhodium phosphine complexes, as well as by palladium PCP pincer complexes, through the spacers of various lengths (C_2_, C_6_, C_12_) (Figure 13) [93,146,147,148,149,335]. Thus, synthesized catalysts were successfully applied in the selective hydrogenation of dienes and polyenes [148], oxidation of cyclic and terminal alkenes [147], hydroformylation [146,335], carbonylation [149,332,334,336,337], and cross-coupling reactions [338,339].

In particular, the total yields of alkenes in the hydrogenation of dienes in the presence of Pd phosphine complex catalysts, based on PAMAM dendrons, anchored to silica, reached up to 99% under the ambient conditions (25 °C, 1 atm. of H_2_, MeOH, 0.5–24 h) [148]. Herein, the total selectivity on alkenes as well as catalyst stability and recyclability increased with the increase in the dendrimer generation and spacer length (C_2_, C_6_, C_12_) between end amino and phosphine groups in the ligand (Figure 16, left) [148]. Hydrogenation mainly proceeded via 1,2-addition; nonetheless, the product distribution was strongly dependent on the substrate structure and, as a consequence, on the intermediate stability [148,288,289], and dienes, giving the higher portion of unsubstituted terminal alkenes, tended to undergo further hydrogenation to alkanes, whose portion may reach 1–33% at full conversion of initial diene (Figure 14) [148].

Applied in the oxidation of terminal alkenes to methylketones, using *tert*-butyl hydroperoxide as an oxidant (55 °C, nonane), the catalyst SiO_2_-G0-Pd (i.e., containing only 3-aminopropyl linker without grafted dendron) gave yields of 27–42%, which decreased from 1-octene to 1-decene and 1-tetradecene, and 11% for cyclohexene within 24 h [147]. Herein, the strong dendritic effect was observed. An increase in the spacer length (Figure 16) from C_2_ to C_12_ allowed for a significant increase catalyst activity and made it possible to oxidize even bulky substrates, such as 5-vinyl-2-norbornene and 4-phenyl-1-butene [147]. It should be noted that the efficiency of these hybrid catalysts, based on prolonged dendrons, depended on the dendrimer generation, spacer length, and substrate size simultaneously, and positive dendritic effect was observed.

Rh complex catalysts, based on the same PAMAM dendron containing support, afforded the hydroformylation of various alkenes [146]. Conversions of 95–99% were achieved for various alkenes within 24 h at 65–75 °C, with a syngas pressure of 70 atm. (CO:H_2_ = 1:1) in CH_2_Cl_2_ medium. Herein, linear alkenes, such as 1-octene, tended to mostly give linear aldehyde, whereas styrenes, vinylnaphthalenes, and vinyl acetates (i.e., with substituents, giving −*M* effect to C=C double bond) gave branched products with selectivity often exceeding 90%. SiO_2_-PAMAM-Rh catalysts display a slightly negative dendritic effect (catalysts of G0—G1 generations effectively worked even at room temperature) and were reused several times without a noticeable decrease in activity [146].

Carbonylation of terminal alkenes in the presence of Pd phosphine complex catalysts, decorating the PAMAM dendrons of 0–4th generations, anchored to silica, in the alcohol medium (115 °C, 7 atm. of CO, MeOH, PPh3, *p*-TsOH, THF or toluene) resulted in the ester formation, with a predominance of linear product, even for styrene-type substrates [149]. Herein, this linear/branched ratio improved at catalyst recycling due to the leaching of the adsorbed less hindrance Pd complexes favoring the branched product formation [149].

Using *otho*-allyl- or vinylphenols and anilines as substrates under the similar conditions (100–140 °C, 7–43 atm. of CO, 0–36 atm. of H_2_, toluene, or CH_2_Cl_2_, 1,4-bis(diphenylphosphino)butane (dppb)), it was possible to obtain five-, six- and seven-membered ring lactones and lactams with yields up to 90% (Figure 15a,b) [332,334]. Varying the reaction conditions (temperature, solvent, CO/H_2_ ratio, and pressure) for the materials, modified by Pd PCP pincer complexes (Figure 16, right), one could direct the reaction to the five- or seven-membered ring lactone formation with the selectivity of 82–91%, significantly exceeding that for a homogeneous non-dendritic Pd PCP complex catalyst [334]. Heterogeneous dendritic PCP pincer Pd catalysts had high stability at recycling, extremely increasing from G0 to G1 dendron generation, with the 7-membered lactone as a predominant product even under the conditions for 5-membered lactone formation in the 1st cycle [334].

In the presence of SiO_2_-PAMAM-Gn-(Pd(PR_2_)_2_)_n+1_ (n = 0–3, R = Ph, Cy) catalysts under carbonylation conditions (100 °C, 7 atm. of CO, MeOH, NEt_3_), various iodoarenes were effectively converted to corresponding arylcarboxylates with quantitative yields within 24–36 h [332]. In the case of carbonyls, assuming cycle formation (e.g., dibenzoxazepinone), there was also possible the use of bromoarenes under elevated temperatures (80/120 °C, 7 atm. of CO, NEt(*i*-Pr)_2_, toluene), with yields exceeding 90%, even for dimeric substrates (Figure 15c) [337]. Catalysts, based on dendrons at least of the 1st generation, appeared highly stable under reuse; for substrates, such as 2-(2-iodophenoxy)aniline, the yields maintained to as high as 95% during the 8 cycles [338].

The same type of catalysts showed high activity in the Heck cross-coupling reactions for both iodo- and bromoarenes with styrene, butyl, or methyl acrylate with the predominant formation of *trans*-products (120–140 °C, Na_2_CO_3_, or NaOAc, DMF) [338,340]. The maximum yields of 85–99% were observed for the substrates with electron withdrawing groups (C(=O)OMe, NO_2_)), and were maintained when recycling the catalyst [341,342]. Herein, the SiO_2_-G0-PCP-Pd catalyst appeared more stable when reused [93].

Hagiwara’s group has developed a heterogenous hybrid catalyst, combining properties of both supported the ionic liquid catalyst (SILC) and dendritic catalyst [343]. Terminally alkylated PAMAM dendrons of the 3rd generation were anchored to nano-silica with a mean particle diameter of 10–12 nm, followed by the impregnation of Pd(OAc)_2_, solved in the ionic liquid. Under these conditions Pd complexes were reduced and formed dendron-encapsulated clusters (Figure 17) arranged into the raspberry-shaped superstructures [343].

Thus, the synthesized Pd-containing catalyst was successfully applied in the Suzuki cross-coupling reactions, using various bromo/iodoarenes and arylboronic acids as substrates under mild conditions (25–60 °C, K_2_CO_3_, H_2_O/EtOH). High biarene yields were favored by the use of phenylboronic acids with donor substituents and arylbromides or aryliodides with acceptor ones, and the quantitative yields for 4-bromoacetophenone, coupled with phenyl- or 1-naphthylboronic acid were achieved already within 30 min at room temperature [343]. It was established that the gradual decrease in activity of the hybrid dendritic SILC catalyst at recycling originated from Pd cluster aggregation, not from Pd leaching [343].

Li and Xu proposed a heterogeneous catalyst for Suzuki and Heck cross-coupling reactions based on poly(ethylene imine) (PEI) dendrons anchored to silica [341]. The latter was activated by (3-chloropropyl)trimethoxysilane grafting, followed by end-Cl atom replacement with diethanolamine according to the *S_N_2* mechanism (Figure 16). The material obtained, containing OH-terminated dendrons of the 1st generation, was further treated with thionyl chloride to replace terminal OH groups by Cl, and then the procedure was repeated until Cl-terminated dendrons of the 2.5th generation were propagated and then capped with 1,4-diazabicyclooctane (DABCO) (Figure 16). Thus, the synthesized carrier, coated by grafted DABCO-terminated dendrons of the 3rd generation, was impregnated with PdCl_2_ solution in EtOH followed by a reduction in the presence of hydrazine in situ, resulting in Pd nanoparticles of 3–10 nm in diameter (Figure 16) [341].

The synthesized SiO_2_-PEI-G3-DABCO-Pd catalyst revealed high activity Suzuki (for *p*-bromoanisole and phenylboronic acid) and Heck cross-coupling reactions (for *p*-iodo- or bromoanisole and styrene). By carefully adjusting the reaction conditions (80 °C, 0.5 h, K_2_CO_3_, EtOH/H_2_O for Suzuki cross-coupling, and 130 °C, 1–3 h, K_3_PO_4_, DMF/H_2_O for Heck cross-coupling), it was possible to achieve the quantitative yields for desired products (Figure 17) [341]. It was established that under the said conditions, yields of 95–99% within 0.5 h in the Suzuki cross-coupling reactions in the presence of the SiO_2_-PEI-G3-DABCO-Pd catalyst were maintained when bromoarenes with *para*- and *meta*-, donor and acceptor substituents, as well as phenylboronic acid with donor *para*-substituents, were used as substrates. Replacing bromoarene for chloroarene resulted in the essential downfall in the catalyst activity; nonetheless, the yields of 77–89% were obtained within 24 h at 130 °C and increased catalyst loading for chloroarenes with acceptor *para*-substituents (NO_2_, CN) [341].

Heck coupling proceeded with quantitative yields within 1 h only for *para*-substituted iodoarenes and styrene. The replacement of iodoarene by bromoarene and, moreover, for chloroarene resulted in the noticeable decrease in the catalyst activity, making the reaction strongly sensitive to substrate structure and electronic properties [344]. The yields of 60–89% within 2–5 h appeared to be achievable for bromo- and chloroarenes only with acceptor *para*-substituents (NO_2_, C(=O)OCH_3_), while the use of donor- or meta-substituted bromoarenes (bromobenzene, 4-bromotoluene, and anisole, 3-bromonitrobenzene) led to the sharp downfall in conversion to 0–25%. The use of acrylic acid, such as alkene instead of styrene, resulted in the yields of 83–84%, achievable only after 10 h both for iodo- and bromoarenes, which may be connected with a pH decrease in the reaction medium [341]. The hybrid SiO_2_-PEI-G3-DABCO-Pd catalyst appeared to be easily recoverable from the reaction mixture. Nonetheless, its activity noticeably decreased in the recycling of Suzuki coupling, which may be attributed to the gradual Pd leaching and/or Pd nanoparticle agglomeration [341].

Mohammadpoor-Baltork and Mirkhani developed Pd nanoparticles, immobilized on nano-silica (10–40 nm), coated with triazine dendritic polymer (*n*STDP) [342,345]. The latter was propagated by the repeated alternating stages of triazine trichloride (cyanuric chloride) and bis(3-aminopropyl)amine addition according to the Csp^2^ *S_N_2* mechanism, until G3 dendrons were obtained (Figure 18). The mean particle size of the further deposited Pd nanoparticles was found to be 3.1 nm [342,345].

Thus, the synthesized hybrid Pd-*n*STDP catalyst was studied in Heck and Suzuki C–C cross-coupling reactions (Figure 19) [342], as well as in C–S cross-coupling reactions between various arylhalides and diaryldisulfides under both microwave (MW) and conventional thermal conditions (Figure 20) [345]. Suzuki coupling proceeded with near quantitative yields (93–96%) within 2–3 h or 6–10 h at room temperature for iodoarenes or bromoarenes, respectively (K_2_CO_3_ as a base, DMF/H_2_O (1:3) as a reaction medium) (Figure 19) [342]; the use of MW irradiation (200 W, 70 °C) allowed the achievement of the same yields after 2–5 min, both for iodoarenes and bromoarenes. Chloroarenes required much longer reaction times (18–24 h; 7–10 min under MW) or more severe reaction conditions (80 °C). Heck coupling between various arylhalides and substituted styrenes required 9–20 h for the near-to-quantitative yields (90–95%) to be reached under the conventional thermal conditions (85 °C, K_2_CO_3_, DMF/H_2_O (1:3)), and 8–20 min under MW conditions (200 W, 70 °C), depending on the halide nature [342]. The presence of electron-withdrawing or, especially, of electron-donating groups in the structure of both substrates resulted in the noticeable decrease in the reaction rate, as in Suzuki and as in Heck coupling reactions [342].

In C–S coupling (Figure 20), yields of 90–95% were achieved within 5 h under conventional thermal conditions (80 °C, Bu_4_NOH, DMF/H_2_O (1:1)) or within 10–15 min under microwave (MW) conditions (Figure 20) with the TOF values, reaching, in the last case, up to 5700 h^−1^, enormously exceeding those for conventional homogeneous and heterogeneous catalytic systems, such as PdCl_2_(dppf)/Zn, CuFe_2_O_4_ or Cu_2_S/Fe [345]. It was found that electron withdrawing *para*-substituents favored the better reaction yields; nonetheless, it was noticeable only for chloroarenes, converted to the corresponding diarylsulfides with yields of 90–90% after 10 h. Moreover, double C–S coupling was successfully conducted with the various *para*-and *meta*-dibromoarenes (including 2,6-dibromopyridine) and diaryldisulfides (including 2,2′-dithiobis(benzothiazole)) under said conditions. The yields appeared as high as 75–95% within 6–10 h under conventional thermal heating and within 10–45 min under MW irradiation. As in the case of mono C–S coupling, the reaction rate and arene disulfide yields were mainly enhanced by electron withdrawing groups (EWG: phenyl, benzo, C=N/pyridine) in the substrates [345]. Hence, using the high activity of a Pd-*n*STDP catalyst even for chloroarenes, especially under the MW conditions, it was possible to synthesize via C–C or C–S coupling reactions with banana- and star-shaped molecules with benzene, pyridine, or triazine cores [342,345].

The hybrid dendritic Pd-*n*STDP catalyst appeared highly stable at recycling, and no decrease in the reaction rate and product yield, nanoparticle agglomeration, or noticeable Pd leaching were observed after seven runs in both C–C and C–S coupling reactions under both thermal and MW conditions [342,345]. A reaction mechanism was proposed, and accordingly, dendron moieties functioned as micro/nanoreactors for Pd nanoparticles, strongly retaining them and, therefore, prevented their leaching (Figure 21) [344,345,346,347,348]. Herein, at the first stage, an oxidative addition of arylhalide (ArX) to the surface of Pd nanoparticle took place, followed by diaryldisulfide addition and halide release in the presence of base. Re-coordination of chemisorbed aryl and aromatic sulfide resulted in the reductive elimination of the end product, diarylsulfide, and the subsequent turn-back of Pd species to the next cycle (Figure 21) [345].

It is important to note that dendron grafting on the silica surface is sufficiently intricate, with the increase in the dendron generation and steric hindrances arising from this and resulting in the incomplete Michael addition and amidation [92,146,252,339,349,350]. Herein, the yields of G3–G4 and higher dendron grafting may not exceed 40–60% [147,252,339,350]. In order to improve the performance of silica-supported dendritic catalysts, it was suggested by Alper and Cao to attach future dendron focal points to the inner walls of the ordered mesoporous materials, such as pore-expanded MCM-41 [351] and SBA-15 (Figure 22) [352,353].

In particular, it was established that the yield of G3–G4 PAMAM dendrons (up to 4.5 m in diameter [85,114,354,355,356]), anchored to pore-expanded MCM-41 material with channel diameter of 10.6 nm, reached 70–75% [351], which was also due to the lower initial APTES loading in comparison with the grafting to amorphous silica [147,339,350], hence resulting in the lower dendron density and less steric hindrances. Analogously covalent immobilization of G3–G4 PAMAM dendrons in the pores of SBA-15 material (pore diameter of ~7.5 nm) proceeded with the yield of ~60% [350].

The last material was further used as a carrier for monometallic Pt, Pd, Au, and bimetallic PdAu and PdPt nanoparticles, resulting in the catalysts for hydrogenation of C=C double bonds [353], reduction of *p*-nitrophenol [354], oxidation of alcohols [355], and Suzuki cross-coupling [357]. Herein, the presence of covalently grafted dendrons provided uniform and narrow well-dispersed particle size distribution (1–3 nm) inside the SBA-15 channels [352,353,357]. Thus, Pd-containing catalysts with the particle size of 1.2–3.4 nm, based on the dendritic organic-inorganic hybrid composites, revealed extremely high activity in the hydrogenation of allyl alcohol under the ambient conditions (30 °C, 1 atm. of H_2_, MeOH), reaching conversions of more than 99% and TOF values of 710–2270 h^−1^ within 20–120 min [353]. A noticeable negative dendritic effect was observed, appearing as a decrease in the reaction rate for higher generation catalysts; nonetheless, the catalyst based on dendrons of the 3^r^ generation mostly suppressed isomerization of allyl alcohol to acetone and demonstrated the best selectivity on 1-propanol, reaching up to 95% [353]. Moreover, the catalyst SBA-15-PAMAM-G3-Pd revealed the highest stability at recycling and storage and was successfully reused several times without any loss of activity, Pd leaching, or nanoparticle agglomeration observed [353].

A similar approach can be applied for the synthesis of G4-PAMAM dendron-encapsulated monometallic Pd and bimetallic PdAu nanoparticles, immobilized in the SBA-15 channels by co-complexation method [357]. These dendrimer-encapsulated PdAu nanoparticles effectively catalyzed Suzuki cross-coupling between various arylbromides and phenylboronic acid under the microwave conditions (100 °C, 30 min, K_3_PO_4_, H_2_O/EtOH, MW), with the yields significantly exceeding those for monometallic Pd catalysts, active mostly in the iodoarene coupling. This phenomenon was attributed to Au^3+^ ions weakening the coordination interactions between Pd^2+^ and dendron amino groups, hence facilitating the reduction of Pd^2+^ to Pd^0^ [357]. The maximum yields (90–99%) for both Pd and PdAu catalysts were achieved using unsubstituted substrates or *para*-substituted phenylboronic acids with electron donating groups (CH_3_, OCH_3_) and *para*-substituted arylhalides with electron withdrawing groups (NO_2_). Herein, the use of sterically hindered *ortho*-substituted phenylboronic acids, even coupled with iodebenzene, resulted in a drastic decrease in the reaction yields down to 30% due to the negative dendritic effect [358]. It was discovered that the mean particle size, arising from the initial metal/dendrimer ratio, had an essential effect on the reaction rate and cross-coupling yield, and the best activity and stability at recycling were revealed for tiny Pd_10_ and Pd_5_Au_5_ nanoparticles of 1.4–1.6 nm in diameter strongly encapsulated inside the dendron cavities [357].

Monometallic Au nanoparticles, encapsulated in PAMAM dendrons of G1–G4 generations, grafted to SBA-15 ordered mesoposous silica, effectively catalyzed the oxidation of secondary alcohols at room temperature and the ambient oxygen pressure in biphasic CH_2_Cl_2_/H_2_O reaction medium, in the presence of K_3_PO_4_ as a base [355]. An increase in the dendron generation from G1 to G4 was found to increase mean particle size from 1.6 to 3.4 nm, which authors explained in terms of steric crowding on the higher generation dendron periphery, interfering the Au^3+^ ions to penetrate inside the dendron cavities, thus resulting in the outer dendron-stabilized nanoparticles [355]. The reaction rate and the yield of corresponding ketone decreased with an increase in the dendrimer generation and increased with a decrease in mean particle size arising from the Au/dendrimer ratio. In particular, the yield of 99% for acetophenone in the presence of the SBA-15-PAMAM-G4-Au_100_ catalyst was attained only within 24 h. Rather, it decreased down to 80–90% for bulky or negatively induced substrates, such as 1-(2-naphthyl)ethanol or 1-(4-chlorophenyl)ethanol, which are able to competitively adsorb via their substituents, or, especially, replace aromatic by alicyclic substrates, such as cyclopentanol or cyclohexanol (50–70%); for primary linear aliphatic C_6_–C_8_ alcohols the conversions did not exceed 4–20% [353]. Nonetheless, in the oxidation of aromatic alcohols, the SBA-15-PAMAM-G4-Au_100_ catalyst can be recycled several times without noticeable loss in activity—in contrast to Au nanoparticles, which are supported on SBA-15 without a grafted dendrimer ligand [353].

With respect to heterogenized G4-PAMAM, dendron-encapsulated Pt nanoparticles, it was discovered that increase in mean particle size (at elevated Pt loadings) resulted in the decrease in the reaction rate both for *p*-nitrophenol and K_3_[Fe(CN)_6_] reduction (NaBH_4_ or Na_2_S_2_O_3_ as a reducing agent, 25 or 40 °C, H_2_O, 30 or 90 min, respectively) [354]. It was presumably due to larger Pt nanoparticles blocking the channels of SBA-15, hence reducing the catalyst-specific surface area and retarding the mass transfer in the reaction medium [354]. Nonetheless, in contrast to SBA-15-PAMAM-Pd catalysts [354], the analogous Pt-containing catalyst appeared did not seem very stable at reuse, maintaining about 50% of initial activity to the fifth cycle due to mechanical losses [354].

Shantz et al. developed monometallic Cu and bimetallic CuAu catalysts, based on Simanek-type melamine-piperidine dendrons, covalently grafted inside SBA-15 channels (Figure 23) [233]. Herein, in spite of the fact, that bimetallic catalyst was synthesized by co-complexation method, Au core–Cu shell nanoparticles were obtained. The catalysts obtained were studied in the alkyne-azide click cycloaddition (50 °C, NEt_3_, THF), and strong negative dendritic effect was observed—also, in comparison with a conventional heterogeneous catalyst Cu/SBA-15 without a dendrimer. The yields of 94% and 81% were reached for Cu/SBA-15 and Cu@G2/SBA-15 after 5 and 24 h, respectively; nonetheless, the presence of dendrimer was found to strongly suppress metal leaching at recycling. The presence of gold in the nanoparticle structure provided a remarkable increase in the catalyst activity: a 100% conversion Cu@G2/SBA-15 catalyst was achieved after 15 h with the reaction rate, quite similar for that of the Cu/SBA-15 catalyst.

By means of IR spectroscopy, it was established that alkyne adsorbed rapidly on the copper surface without any induction period before azide adsorption, thus assuming the linear intermediate formation, antecedently the ring closure and triazole formation, to be the rate-limiting step in the reaction (Figure 24) [233]. Herein, all intermediates observed were the same, as in homogeneously catalyzed alkyne-azide cycloaddition. No electron synergistic effect between copper and gold was revealed, indicating that the increase in activity for the bimetallic catalyst was mainly due to enhanced copper atom distribution on the nanoparticle surface [233].

#### 3.1.2. Divergent Dendrimer Grafting on Polymer and Carbon Supports

Linear insoluble polymers, such as polystyrene (PS), are another common support for dendron grafting, thus resulting in dendronized polymers (DenPols), which can be synthesized in different ways depending on the dendrimer’s nature [63,72,75,86,93,358,359,360,361]. In particular, to produce a focal point of future PAMAM dendrons supported on polystyrene, the latter is preliminarly activated by a chlorometylation reaction followed by substitution for the amino group directly according to the *S_N_2* mechanism [362,363] or via phthalimide stage (Figure 25) [364,365,366,367].

Thus, Feng et al. studied the hydrogenation of 5-membered heterocycles, such as furan, pyrrole, and thiophene, in the presence of Pd catalysts, based on salicyliden-terminated PAMAM dendrons of the 2nd generation (similar to those in Figure 12), anchored to polystyrene (PS), at an ambient temperature and pressure in the ethanol medium [368]. Quantitative conversions were achieved for all substrates, and products of ring opening and C–C bond cleavage were also obtained. The PS-PAMAM-Pd catalyst was successfully recycled 10 times with a gradual decrease in activity [368].

Krishnan and Sreekumar used polystyrene-supported PAMAM dendrons of various generations for the synthesis of encapsulated Pd nanoparticles, followed by their application in the Suzuki cross-coupling (100 °C, Na_2_CO_3_, 1,4-dioxane/water) [364]. Herein, Pd nanoparticles, stabilized by G1 and G2 dendritic polymers, tended towards agglomeration, whereas G3 support provided well-separated nanoparticles of 10–20 nm in diameter (Figure 18). In the presence of a catalyst, based on dendrons of the 3rd generation, the reaction between phenylboronic acid and 4-nitroiodobenzene proceeded with the yield of 90–95% within 12–24 h. It was established that the use of substrates with donor substituents (CH_3_, C(CH_3_)_3_, NH_2_) both for arylboronic acid and, especially, for arylhalide resulted in the significant decrease in the catalyst activity and reaction yield, which was extremely noticeable for arylbromides and arylchlorides. PS-PAMAM-G3-Pd had high stability at recycling, and only a slight decrease in the reaction yields was observed during 6 cycles due to Pd leaching [364].

The analogously hybrid metal complex catalyst Mn (II), based on PS-PAMAM-G3 dendritic carrier, demonstrated a high efficiency in the oxidation of various secondary alcohols, giving near-to-quantitative conversions (94–98%) within 3 h at room temperature, using K_2_Cr_2_O_7_ or KMnO_4_ as oxidizers and acetone as solvent [252]. PS-PAMAM-G3 material itself effectively catalyzed Knoevenagel condensation, giving yields of 95–100% within 10–120 min at 30–50 °C, using ethanol as solvent [366]. Herein, dendritic material acted as a base catalyst, and the reaction rate decreased with the increase of both carbonyl and the active methylene compound [366]. For both above-mentioned materials, a strong, positive, dendritic affect and excellent recyclability were revealed [252,366]. An increase in the degree of the polystyrene cross-linking furthered a slight conversion decrease [252], whereas an increase in the dendronization degree, in comparison, positively influenced the catalyst activity [366].

Jiang, Yang, and Pan suggested a novel approach for the synthesis of gold nanoparticles, stabilized by PAMAM dendrons divergently grafted to polystyrene beads [367]. This approach included preliminary protonation of alkylated tertiary dendron end groups, followed by complex formation with NaBH_4_; this complex functioned as a reducing agent for KAuCl_4_^–^, added after all the others, thus resulting in nanoparticles of 5–10 nm in diameter (Figure 26).

Thus, synthesized polymeric catalysts were studied in the oxidation of primary and secondary benzyl alcohols and in arylboronic acid homocoupling (Figure 27). As for primary benzyl alcohols, the yield of corresponding benzaldehydes reached 99% within 16 h for substrates with +*I* or +*M para*-substituents (Me, OMe, Cl, Br, *i*Pr) at room temperature and ambient oxygen pressure in the presence of the PS-PAMAM-G3-Au catalyst [367]. For the secondary alcohols, the quantitative conversions to corresponding methylketones were achieved only within 32 h, all other conditions being equal. Replacement of the substituent in *para*-position with the hydroxymethyl group by the NO_2_ group’s strong −*M*-effect resulted in a drastic decrease in the reaction yield down to 13%.

Additionally, it should be noted that despite the PS-PAMAM-G3-Au catalyst exhibiting the best efficiency, for all other catalysts, based on lower generation dendrons, a strong negative dendritic effect was observed, and the yields of benzaldehyde reached 73%, 50%, and 39% for G0, G1, and G2 catalysts, respectively [367]. Homocoupling of phenylboronic acids in the presence of the PS-PAMAM-G3-Au catalyst proceeded with the yield of 94–99% within 24 h at room temperature, air subjected, in water medium, using K_2_CO_3_ as a base. Herein, biphenyls were selectively produced from the substrates with donor *meta*- and *para*-substituents (Me, OMe), whereas ortho-substituted substrates resulted in hydrolysis product (phenol) [367]. The catalyst PS-PAMAM-G3-Au appeared very stable at reuse and maintained a yield in benzyl alcohol oxidation as high as 99% after 16 h and 52–55% after 5 h during 14 cycles [367].

Arya, Alper, and Manzer designed polystyrene-supported poly(aryl amide) dendrons in a divergent route, using a peptide synthesis approach (Figure 28), and then modified them by Rh phosphine complexes [369]. The synthesized heterogeneous Rh complex catalysts were applied in the hydroformylation of various olefins at 65 °C and 140 atm. of syngas (CO:H_2_ = 1:1). The positive dendritic effect was discovered, and quantitative conversions were achieved within 24 h for G2–G3 catalysts, which did not lose their efficacy for 4–6 cycles at reuse. Herein, the portion of branched product often exceeded 90% not only for styrene-type substrates, but also for vinyl acetate [369].

Dahan and Portnoy suggested materials, based on Fréchet type poly(aryl ether) dendrons, immobilized on hydroxylated polystyrene (Wang resin) [370,371]. The dendron grafting was proposed here in two different ways (Figure 29) [370,371]. The first one suggested the attachment of 5-hydroxyisophthalate moiety to hydroxylated polystyrene via a Mitsunobu reaction in the presence of diisopropyl azodicarboxylate (DIAD) and triphenylphosphine-sulfonamide betaine in CH_2_Cl_2_ medium, followed by reduction using a LiBH_4_ reaction in the presence of B(OMe)_3_ in THF medium (Figure 29a) [372]. According to the second approach, 3,5-bis(acetoxymethyl)-phenol unit was grafted to Wand resin via a Mitsunobu reaction again in the presence of PPh_3_ and DIAD in THF medium, followed by hydrolysis of ester groups in the presence of tetrabutylammonium hydroxide (TBAH) in water/THF medium (Figure 29b) [373]. Thus, hydroxymethyl-terminated poly(aryl ether) dendrons of various generations were obtained; their subsequent modification by diphenylphosphinobenzoic acid in the presence of diisopropylcarbodiimide (DIC) in CH_2_Cl_2_ (for *ortho*-isomer) or DMF medium (for *para*-isomer), catalyzed by 4-(*N,N*-dimethylamino)pyridine (DMAP), resulted in the heterogenized diphenylphosphine-terminated ligands for metal complex catalysis [370,374].

In particular, Co-containing catalysts proved their efficacy in Pauson–Khand [2+2+1] cycloaddition of alkyne, alkene, and carbon monoxide (Figure 30) [374], whereas Pd-containing catalysts appeared highly active in the Heck cross-coupling reaction (Figure 30) [370]. A positive dendritic effect was found out in both processes, which was explained in terms of a decrease in the resin internal coordination cross-linking, an increase in the number of catalytic sites on the polymer surface, and enhanced stability of the metal nanoparticles formed in situ when the dendron generation increased [370,374]. The best yields in Pauson–Khand cycloaddition were as high as 68% within 24 h at 70 °C and an ambient CO pressure in THF medium, using Co-containing catalysts with *o*-diphenylphosphinobenzoyl dendron end group moieties, which authors attributed to the Co bidentante coordination [374].

In the Heck coupling between bromobenzene and methyl acrylate (Figure 31), the quantitative yield of methyl *trans*-cinnamate was analogously obtained for the catalyst, based on dendrons of the 3rd generation, decorated on periphery by Pd complexes with *p*-diphenylphosphinobenzoyl fragments, rather than exceeding that for the homogeneous catalyst under the same conditions (80 °C, NEt_3_, NMP, 72 h) [370]. Herein, milder reaction conditions (80 °C, 72 h vs. 120 °C, 14 h) provided much higher selectivity on cross-coupling products in comparison with bromobenzene homocoupling (200:1 vs. 32:1). Replacement of methyl acrylate by styrene, possessing less −*M*-effect, required more severe reaction conditions for better conversion and selectivity, whereas +*M* butyl vinyl ether resulted in a mixture of products of both α- (α-butoxystyrene and acetophenone) and β-arylation (*cis*- and *trans*-β-butoxystyrenes), and gradually increased together with the increase in the dendron generation along with the β/α ratio, which decreased. The phenomenon observed was explained in terms of the higher portion of the olefin insertion proceeding through cationic intermediates, arising from +*M*-effect of the OBu group and increasing in the polar environment, provided by the catalytic species of a higher generation [370].

Other interesting carbon materials for dendron immobilization are carbon nanotubes (CNTs) [372] and polyaromatic frameworks (PAFs) [363]. The latter (otherwise called porous aromatic frameworks) are three-dimensional polymers, who’s rigid, diamond-like structure, consisting of tetrahedral node atoms connected with polyphenylene edges (Figure 19), provides high specific surface area and thermal stability [375,376,377,378]. Earlier it was demonstrated that Pd nanoparticles, immobilized on polyaromatic frameworks, functionalized with amino groups, appeared noticeably inferior to those based on unmodified PAFs, as catalysts for hydrogenation of various unsaturated compounds (phenylacetylene and styrene, linear alkenes, alkynes, and dienes) [379]. Nonetheless, this Pd/PAF-NH_2_ catalyst exhibited excellent stability, maintaining a conversion of about 30% in the hydrogenation of phenylacetylene during 6 cycles [379]. Herein, amino groups attached to polyphenylene edges of PAFs can be considered as G0 dendrons.

The dendron grafting to PAF was performed via chloromethylation stage followed by replacement with diethanolamine according to S*_N_2* mechanism (Figure 25) [363]. Thus, polyaromatic frameworks were obtained, coated in OH-terminated PEI dendrons of the 0th generation. Subsequent chlorination with thionyl chloride and replacement with diethanolamine again gave materials with dendrons of the 1st generation (Figure 32) [363]. The deposition of Pd nanoparticles was conducted in a typical way for polyamine dendritic catalysts, including the impregnation with Pd(OAc)_2_ solution in chloroform followed by the reduction in the presence of sodium borohydride in the ethanol/water mixture [363]. It was surprising that more hindered G1 dendrons favored larger particle formation with broader-sized distribution in comparison with less hindered G0 dendrons (4.5–13 nm vs. 1.5–4 nm), which authors attributed to the blocking of pores by branched aminoethanol moieties and, as a consequence, nanoparticle formation outside the pores [363].

The grafting of diethanolamine moieties to a PAF surface significantly enhanced the catalyst performance in the hydrogenation of various unsaturated compounds (cyclic and linear alkenes, alkynes and dienes, phenylacetylene, and styrenes) in comparison with just amino-modified PAF, allowing it to reach TOF values more than 100,000 h^−1^ versus ~55,000 h^−1^ for material based on PAF-20 with the terphenylene edges [363,379]. It should be noted that the activity of PAF-based catalysts is strongly dependent on the pore size determined by the number of phenylene rings in the edges [377,378]. As a consequence, change from Pd-PAF-20-PEI-G0 to Pd-PAF-30-PEI-G0 with quarterphenylene edges resulted in the increase in the catalyst-specific activity up to 300,000 h^−1^ for linear alkynes and dienes (Figure 19) [363]. Dendron grafting and further propagation decreased the pore volume and specific surface area; as a consequence, the specific catalyst activity decreased again, and conversions often did not exceed 1–5%. Nonetheless, for 1-hexyne the TOF values of as high as 100,000 h^−1^ and 190,000 h^−1^ for Pd-PAF-20-PEI-G1 and Pd-PAF-30-PEI-G1, respectively, were achieved (Figure 20) [363]. Similar to Pd-PAF-NH_2_ [380], Pd-PAF-PEI catalysts proved their stability at recycling. An observed decrease in activity at the first two cycles, especially noticeable for Pd-PAF-20-PEI-G0, was attributed to the leaching of Pd nanoparticles formed outside the pores [363].

With regards to carbon nanotubes, their functionalization is mainly performed via oxidazing treatment by an HNO_3_/H_2_SO_4_ mixture, causing structure defects on the surface (C(O)OH, C=O and OH groups, arising from C=C and C–C bond cleavage) [372]. In particular, by the esterification of the surface carboxyl groups, arising from the defects on the outer carbon nanotube walls, with propargyl alcohol followed by the cycloaddition of 3-azidopropylamine in the presence of Cu(OAc)_2_ and sodium ascorbate as catalysts, the focal points (G0) for the subsequent PAMAM dendron grafting were produced (Figure 33) [372]. The latter was conducted by the divergent method, using the repeated stages of methyl acrylate Michael addition and the subsequent amidation by ethylene diamine (Figure 33). Hence the materials, based on multi-wall carbon nanotubes coated with dendrons of up to the 3rd generation (PAMAM-g-MWCNTs), were obtained and used for Pd nanoparticles deposition analogously to the homogeneous PAMAM dendrimer-based catalysts [115,372]. Mean particle size was found to be 1.8–2.8 nm for Pd-G3-PAMAM-g-MWCNTs material.

The synthesized Pd-PAMAM-g-MWCNTs catalysts were examined in the Heck cross-coupling reaction, and a strong positive dendritic effect was revealed, which authors explained in terms of higher Pd nanoparticle stability and leaching retarding in the presence of dendrimers of higher generations [372]. Yields of 95–99% were achieved already after 1–6 h at 100 °C, with the presence of K_2_CO_3_ as a base in the NMP medium, even for chloroarenes. Herein, electron withdrawing substituents (such as NO_2_ or C(=O)OMe) in both arylhalide and alkene favored the improved catalytic activity and, therefore, to the maximum reaction yields. Pd-G3-PAMAM-g-MWCNTs revealed the best stability at recycling, and no decrease in the catalyst performance was observed during six cycles [372].

### 3.2. Convergent Dendrimer Grafting

The convergent approach, implying the covalent attachment of the already synthesized dendrimers or dendrons to the preliminary functionalized carrier, escapes the defects resulting from irregular dendrimer coatings, which are inevitable for multi-step divergent synthesis due to diffusion limitations increasing with the increase in the dendron generation [92,146,147,252,339,349,350,364,368,381]. The anchoring of readily prepared dendrimers of the appropriate generation can be performed in one stage, thus providing an atom-economical strategy and, as a consequence, much higher yields of the dendronized hybrid support [382].

#### 3.2.1. Convergent Dendrimer Grafting to Carbon-Based Supports

Giacalone, Gruttadauria, and Prato have developed materials, based on PAMAM dendrons of the 2.5th and 3rd generations, covalently attached to single wall carbon nanotubes (SWCNT) in one step (Figure 34) [10,64,382]. Initial PAMAM dendrimers with a disulfide core underwent radical cleavage on S–S bonds, easily grafting after to non-functionalized pristine nanotubes [380]. The subsequent treatment with Na_2_PdCl_4_ aqueous solution followed by NaBH_4_ reduction resulted in well dispersed palladium nanoparticles with mean sizes of 3.4 and 1.6 nm for G2.5- and G3-based catalysts, respectively.

The catalysts obtained, based on dendrons of the 3rd generation, revealed extremely high activity in Suzuki and Heck cross-coupling reactions, providing quantitative yields within 5–22 h under moderate conditions (Suzuki coupling: ArBr, Ar’B(OH)_2_, K_2_CO_3_, EtOH/H_2_O (1:1), 50 °C, 5–22 h; Heck coupling: ArI, H_2_C=CH–C(=O)OMe, NEt_3_, DMF/H_2_O (4:1), 90 °C, 5–16 h) (Figure 35) [382]. Herein, in Suzuki coupling, the maximum reaction rate and highest product yields were observed for unsubstituted phenylboronic acid and arylbromides with EWG substituents in para-position; moreover, a SWCNT-PAMAM-G3-Pd catalyst appeared to be able to promote coupling for aryldibromides and even for arylchlorides. Heck coupling of methyl acrylate in the presence of a SWCNT-PAMAM-G3-Pd catalyst proceeded similarly for aryliodides with different substituents (in para and meta positions, +*I*, +*M*, –*M*), giving yields of 96–99% within 5 h—except 4-nitrophenyliodide, for which an 85% yield was achieved only after 16 h.

The catalyst SWCNT-PAMAM-G2.5-Pd, based on the ester-terminated dendrons of the 2nd generation, appeared much less active and much more sensitive to substrate structure and electron properties, providing the yields of 97–97% only after 22 h, which authors attributed to the worst stabilization of Pd nanoparticles by ester groups in comparison with amino groups [382]. Under microwave conditions at 120 °C in the presence of the SWCNT-PAMAM-G3-Pd catalyst, it was possible to reach the quantitative yields already within 5 min, thus providing TOF values up to 550,000 h^−1^. This extremely high catalyst activity was explained in terms of the “release and catch” mechanism (Figure 36) [382,383,384] according to that Pd species, interacting with substrate molecules, leached from the nanoparticle surface, and, therefore, the catalytic cycle partly proceeded in the homogeneous phase; on the other hand, the released Pd species can be captured by the dendron moieties, forming nanoparticles again, but of a larger size (3.5 nm). Nonetheless, in spite of partial Pd leaching in the process, the catalyst SWCNT-PAMAM-G3-Pd could be easily recovered and maintain its efficiency in cross-coupling reactions, thus providing yields as high as 99% during 6 cycles at least [382].

Earlier, the anchoring of poly(aryl ether) Fréchet type dendrons to preliminary oxidized multiwall carbon nanotubes (Figure 37) [86,385] and poly(ester amide) Newkome type dendrons (Figure 38) to single-wall carbon nanotubes [25,62,386] were described. In the last case, the material obtained was used for immobilization of CdS nanoparticles as small as 1.4 nm in diameter to develop very stable quantum dots (QD), maintaining their UV absorption behavior more than 3 months—in contrast to CdS nanoparticles, stabilized by ungrafted dendrons and, therefore, tend towards aggregation and precipitation [25,386,387,388].

Bimetallic PtRu nanoparticles of 2.6 nm in diameter, prepared by co-complexation method and encapsulated into PAMAM dendrimers covalently grafted to oxidized carbon nanofiber (CNF, Figure 39), proved their efficacy in the electrocatalytic methanol oxidation, demonstrating a sufficiently better performance in comparison with commercial PtRu/C catalyst [389].

Another strategy was developed for the anchoring of PPI dendrimers to the inner walls of the ordered mesoporous resins (OMR) [362,390]. The latter are carbon materials of SBA-15 or MCM-41 type, obtained by a soft-template method from the polymerization of phenol and formaldehyde in the presence of a poly(ethylene glycol)-block-poly(propylene glycol)-block-poly(ethylene glycol) template (Pluronic P123 or F127) [391]. Anchoring was conducted through the chloromethylation stage for the activation of the carrier surface (Figure 40) [362,390]. Further deposition of Pd nanoparticles was performed in two-stage method, typical for PAMAM and PPI dendrimers and including complexation with Pd^2+^ ions, followed by reduction in the presence of NaBH_4_ [362,390]. The mean size was 3.7 nm, which was consistent with the pore size of MPF-PPI-G3 material (Figure 21) [362,390].

Thus, obtained hybrid OMR-PPI-G3-Pd catalyst revealed a good performance in the hydrogenation of naphthalene (400 °C, 50 atm. of H_2_, 5 h, hexane) [362] and phenylacetylene (80 °C, 10 atm. of H_2_, 15–60 min, solventless) [362], and an excellent one in the hydrogenation of 1-octyne, 4-octyne and 1-hexyne (80 °C, 10 atm. of H_2_, 15–120 min, solventless) [390]. Hydrogenation of octynes proceed with a quantitative conversion already within 15 min, giving TOF values up to 2000 min^−1^ (~120,000 h^−1^) and total selectivity on alkenes of ~98%, with the isomerization portion for octenes not exceeding 10–15% [390]. Replacement of 1-octyne for tinier 1-hexyne resulted in a noticeable decrease in both catalyst activity and selectivity (conversion of 99% within 60 min, total selectivity on alkenes of <90%, TOF ~ 1500 min^−1^/90,840 h^−1^), which authors attributed to the worse adsorption of 1-hexyne [392]. Replacement of linear alkynes by bulky phenylacetylene required 2 h for quantitative conversion (TOF ~1000 min^−1^/61,920 h^−1^), while selectivity on styrene was 92% [362].

Hydrogenation of naphthalene, in the presence of the same catalyst, resulted in the preferential decalin formation with the selectivity of 81% at a conversion of 90% within 5 h (TOF ~ 220 min^−1^/13320 h^−1^) [362]. The major product of guaiacol hydrogenation was 2-methoxycyclohexanol with the selectivity of 90–97% (100 °C, 50 atm. of H_2_, 1–5 h, H_2_O) [393]. Addition of sulfuric acid to the said catalytic system, as well as prolongation of the reaction time, favored the increased portion hydrodeoxygenation products, such as methylcyclohexane, methoxycyclohexane, and cyclohexanol, with the total selectivity being up to 98% at 100% guaiacol conversion, thus significantly exceeding those for a conventional Pd/C catalyst [394]. The OMR-PPI-G3-Pd catalyst did not lose its activity and selectivity at recycling in the 1-octyne hydrogenation [390], maintaining a conversion of more than 95% during 6 runs; however, it appeared unstable under the conditions of guaiacol [394] and, especially, of naphthalene hydrogenation [362].

Another interesting example of a “graft-to” convergent strategy was dendronized polymer material, bearing as side chains the alternating Fréchet type poly(aryl ether) and Newkome type poly(aliphatic ester) dendrons of the 3rd generation [75,392]. The synthesis was performed via orthogonal double-click alkyne-azide cycloaddition and Diels-Alder reaction between poly(aryl ether) and poly(aliphatic ester) dendrons containing alkyne and furan-protected maleimide groups at their focal points, respectively, with linear polystyrene carrying azide and anthracene moieties in alternating blocks (Figure 22) [392].

#### 3.2.2. Convergent Dendrimer Grafting to Silica-Based Supports

Rhee and Chung suggested the grafting of ready synthesized PAMAM dendrimers to amorphous silica via the reaction with (3-glycidoxypropyl)trimethoxysilane (GPTMS), followed by post-modification with chiral moieties on periphery (Figure 41) [92]. The materials obtained exhibited a noticeable negative dendritic effect in the enantioselective addition of diethylzinc to benzaldehyde; conversion and enantiomeric excess did not exceed 80% and 20%, respectively and decreased with the increase in the dendrimer generation [92].

Insertion of spacers between the dendrimer and chiral groups (Figure 41) resulted in a slight increase in conversion and enantiomeric excess due to the better access of substrates to catalytic centers and weakening of multiple interactions among the chiral groups on the dendrimer surface [92]. The maximum conversions (up to 85%) and enantiomeric excess (up to 37%) were found for the catalysts based on dendrimers of the 3rd and 4th generations. However, such catalysts gradually lose their activity at recycling [92].

The similar approach was applied for immobilization of G3 PAMAM dendrimers on the inner surface of titania nanotubes (TiO_2_ NT) [73]. It was discovered that anchoring of PAMAM dendrimers enhanced drug loading capacity and drug release properties of TiO_2_ NT, simultaneously decreasing the cytotoxicity of the latter.

Vallet-Regí et al. offered the immobilization of PPI dendrimers of different generations in the pores of the ordered SBA-15 material, using preliminary dendrimer modification by (3-isocyanatopropyl)triethoxysilane (Figure 42) [74]. The composites synthesized appeared as prospective drug deliverers, with the drug dosage directly controlled by the dendrimer generation via electrostatic and host–guest interactions [74].

Jayaraman and co-workers developed a heterogeneous Pd complex catalyst, based on a phosphine-terminated PETIM dendrimer of the 1st generation, anchored to amorphous silica, modified with (3-chloropropyl)trimethoxysilane (Figure 43) [102]. Thus, a synthesized hybrid SiO_2_-PETIM-PPh_2_-Pd catalyst appeared highly active in the hydrogenation of various olefins under the ambient conditions (25 °C, 1 atm. of H_2_, AcOEt). The yields of 94–100% were achieved within 1–5 h. The catalyst activity and reaction rate were strongly affected by the negative dendritic effect and decreased for sterically hindered substrates, especially with the internal C=C double bonds; maximum reaction rates were observed for styrenes with terminal C=C double bonds, while C=O double bonds and phenyl rings in the substrate structure remained untouched [102]. The catalyst SiO_2_-PETIM-PPh_2_-Pd demonstrated extremely high stability at reuse and storage, maintaining 100% of styrene during 10 reaction cycles (25 °C, 1 atm. of H_2_, 1 h, AcOEt) [102]. Moreover, under the reaction conditions at recycling, full reduction of Pd^2+^ to Pd^0^ took place, and Pd nanoparticles of 2–4 nm in diameter were formed in situ without any Pd leaching or particle agglomeration observed [102].

Maximov and Karakhanov suggested the grafting of PPI dendrimers of the 3rd generation to silica-polyamine composites (SPC) [395] via the Mannich reaction (Figure 44) [171]. Such hybrid organo-inorganic PPI dendrimer-containing material was further applied as a carrier for both mono- and bimetallic nanoparticles [171,393]. Two-step impregnation of Pd(OAc)_2_, followed by reduction in the presence of NaBH_4_, resulted in well-dispersed Pd nanoparticles with a mean particle size of ~2 nm, stabilized by both anchored PPI dendrimers and poly(allyl amine), simultaneously [171].

The synthesized SPC-PPI-G3-Pd catalyst demonstrated very high activity in the hydrogenation of styrene, phenylacetylene, and conjugated dienes (70 °C, 10–30 atm. of H_2_, 15–60 min, benzene or toluene as solvent) with TOF values reaching 100,000–230,000 h^−1^, and thus rather exceeding those for SPC-Pd catalysts without grafted dendrimers in the structure [171]. The separate hydrogenations of styrene and phenylacetylene in the presence of a SPC-PPI-G3-Pd catalyst proceeded with similar rates; nonetheless, by carefully adjusting the reaction conditions (the reaction time, hydrogen pressure, and substrate/Pd ratio) it was possible to selectively convert phenylacetylene to styrene with a yield of ~82%.

The product distribution in the hydrogenation of dienes was significantly dependent on the intermediate stability [288,289] and C=C double bond accessibility in both the substrate and alkenes formed. The total selectivity on alkenes reached 93–99%, even at quantitative and near to quantitative diene conversions and hydrogen pressured of 30 atm. (Figure 45) [171], which rather exceeded that for Alper’s catalysts, based on PAMAM dendrons applied under much milder ambient conditions (25 °C, 1 atm. of H_2_, MeOH) [148]. Herein, the products of hydrogen *trans*-1,4-addition were predominant, in contrast to Pd nanocatalysts based on PAMAM or aryl-cored poly(carbosilane/triazol/aryl ether) dendrimers [148,150]. The SPC-PPI-G3-Pd catalyst proved its recyclability and could be reused several times with a slight decrease in activity due the mechanical losses [171].

Silver addition to palladium, supported on an SPC-PPI-G3 hybrid carrier, by the co-complexation method resulted in a drastic decrease in the catalyst activity and TOF values, not exceeding 30,000 h^−1^, but also in the noticeable increase in the selectivity on styrene in the phenylacetylene hydrogenation (80 °C, 10–30 atm. of H_2_, 15–60 min, solventless) [393]. By carefully adjusting the reaction conditions (the reaction time, hydrogen pressure, substrate/Pd and Pd/Ag ratios) it was possible to achieve a styrene yield as high as 89–93%.

Hydrogenation of 1-octyne in the presence of SPC-PPI-G3-Pd/Ag catalysts under the said conditions was accompanied by the isomerization of C=C double bond in 1-octene formed that was typical for Pd-containing catalysts [116,124]. It should be noted that 1-octyne and resultant octenes easily underwent hydrogenation compared with phenylacetylene, and total yield of octenes did not exceed 83%, with the selectivity on 1-octene not exceeding 77% among the other hydrogenation products [393]. Nonetheless, the hybrid bimetallic dendritic SPC-PPI-G3-Pd/Ag catalysts appeared highly stable and were recycled several times without any loss in activity and selectivity.

Another approach for the synthesis of hybrid organo-silica ready dendrimer-containing materials consisted in the co-hydrolysis in situ of Si(OEt)_4_ with PPI or PAMAM dendrimers, peripherally modified with (3-glycidoxy)propyltrimethoxysilane (Figure 46) [396]. The silica channels were constructed around the dendrimer templates via the sol-gel-method, resulting in microporous material with a pore size of 1–2 nm (Figure 23a). Co-hydrolysis in the presence of additional polymer templates, such as Pluronic P123, favored the formation of SBA-15 type mesoporous material with pore diameters of 5.5–7.5 nm (Figure 47, Figure 23b) [396]. In the presence of the polymer template, but in the absence of Si(OEt)_4_, it was possible to obtain a mesoporous dendrimer network with the silanol groups in the node’s lack of the silica matrix (Figure 48) [193]. It should be noted that, previously, the sol-gel-method via co-hydrolysis in situ was developed for organo-silica materials of SBA-15 and MCM-41 types, containing aromatic [397,398], bipyridyl [399] and chiral tartamide moieties in the matrix structure [400].

Thus, synthesized organo-silica dendrimer-based materials were applied as carriers for Pd and Ru nanoparticles, whose mean size and distribution appeared to be significantly dependent on the carrier structure and dendrimer generation [193,396]. Herein, Ru nanoparticles mostly formed as dendrimer-encapsulated, with the mean particle sizes of 1.0 and 1.3 nm for SiO_2_-PPI-G3-Ru and *meso*-SiO_2_-PPI-G2-Ru, respectively (Figure 24) [396], whereas Pd nanoparticles preferentially formed as dendrimer-stabilized with the mean particle sizes of 2.7–3.3 nm (Figure 25) [193]. Moreover, the shape of particle size distribution for the microporous catalyst SiO_2_-PPI-G3-Pd appeared similar to the pore size distribution of the initial carrier (Figure 25b) [193].

Dendrimer-based organo-silica Pd-containing catalysts were studied in the hydrogenation of various styrenes, alkynes, and dienes (80 °C, 10–30 atm. of H_2_, 15–60 min) [193]. In a whole, the catalytic behavior of *meso*-SiO_2_-PPI-G2-Pd and SiO_2_-PPI-G3-Pd was very similar to that for SPC-PPI-G3-Pd [171], and by carefully adjusting the reaction conditions (the reaction time, hydrogen pressure and substrate/Pd ratio) it was possible to achieve total alkene yields of 83–98% in the hydrogenation of alkynes and dienes [193]. Herein, the mesoporous *meso*-SiO_2_-PPI-G2-Pd catalyst appeared to be the maximum activity in the hydrogenation of styrenes, including bulky and rigid *trans*-stilbene, 4-phenylstyrene, and 1,1-diphenylethylene—in spite of clear-cut negative dendritic effect (Figure 26)—with TOF values exceeding 231,000 h^−1^ for styrene [193], which was similar to those reached in the presence of the SPC-PPI-G3-Pd catalyst [171].

The highest conversions and total alkene selectivities of up to 99% in the hydrogenation of terminal alkynes, tiny and flexible conjugated dienes, as well as of +*I*-substituted 4-*tert*-butylstyrene, were found for the microporous catalyst SiO2-PPI-G3-Pd, based on dendrimers of the 3rd generation and, therefore, providing a larger amount of donor amino groups, increasing electron density on Pd nanoparticles and thus facilitating the desorption of alkene formed [128,171,193,285]. Hence, the competition between steric and electron factors took place, and maximum TOF values reaching up to 400,000 h^−1^ (Figure 27) were achieved for 1-hexyne, isoprene, and 2,4-dimethyl-2,4-hexadiene [193]. The catalyst *meso*-PPI-G2-Si-Pd, obtained without silica matrix formation, was significantly inferior in performance both to *meso*-SiO_2_-PPI-G2-Pd and *meso*-SiO_2_-PPI-G2-Pd; nonetheless, it was the only catalyst providing the *trans*-cyclooctene formation with selectivity of 90–95% [193].

Ru nanoparticles containing catalysts based on PPI dendrimers immobilized in silica pores by co-hydrolysis in situ were studied in the hydrogenation of alkylphenols, levulinic acid, and its esters in water medium [396,401]. The main reaction products of phenol hydrogenation (70–85 °C, 10–30 atm. of H_2_, H_2_O, 0.5–6 h, water) were corresponding cyclohexanols with total yield up to 98% [396]. Herein, *cis*-alkylcyclohexanols were predominant (up to 70% selectivity) at less reaction times (0.5–2 h), especially for bulky and/or *ortho*-substituted phenols (e.g., *o*-allylphenol or *p*-*tert*-butylphenol), while prolongation of reaction time (up to 6 h) resulted in the increased portion of *trans*-alkylcyclohexanols (up to 90%) due to the impact of basic dendrimer amino groups, stabilizing the intermediate enol form and thus facilitating its rotation [396,402,403]. A hydrogenolysis, the typical feature of Ru-based catalysts [1], was observed; its portion reached 20–30% among the reaction products, gradually decreasing from cresols to *p*-*tert*-butylphenol [396].

Hydrogenation of levulinic acid and its esters mostly proceeded through the preferential cyclization to intermediate angelica-lactones, followed by reduction to γ-valerolactone (GVL) (Figure 49), which was the major and desired reaction product (80–120 °C, 30 atm. of H_2_, H_2_O, 2–6 h, water) [401]. Basic PPI dendrimer amino groups, being able to deprotonate the levulinic acid molecules or to catalyze levulinic ester hydrolysis, were assumed to be responsible for the said reaction pathway (Figure 50). Hence, for the SiO_2_-PPI-G3-Ru catalyst, based on dendrimers of the 3rd generation with the larger amount of basic amino groups, the selectivity on γ-valerolactone exceeded 90% even at conversions of 10–50% [401].

For hydrogenation of both phenols and levulinic acid and its esters, the strong negative dendritic effect and competition between steric and electron factors took place, and mesoporous and a sterically less hindered *meso*-SiO_2_-PPI-G2-Ru catalyst appeared much more active than a microporous, sterically more hindered SiO_2_-PPI-G3-Ru catalyst, reaching TOF values up to 6000 h^−1^ versus 1800 h^−1^ for phenols [396] and 1830 h^−1^ versus 930 h^−1^ for levulinic acid and its esters [401]. The increase in the substrate size resulted, on the one hand, in the respective decrease in activity for both catalysts, and on the other hand, it was surpassed by the sufficient +*I*-effect of substituent in the substrate, such as in *o*-allylphenol, *p*-*tert*-butylphenol, or ethyl levulinate, resulting in the increased activity for both catalysts (Figure 28) [396,401].

Dendrimer-based organo-silica was easily recoverable and proved their recyclability in the repeated hydrogenation of various substrates (styrene, 1,3-cyclohexadiene, and 2,5-dimethyl-2,4-hexadiene for Pd catalysts [193]; phenol and levulinic acid for Ru catalysts [396,401]) with just a slight decrease in activity due to the mechanical losses.

## 4. Dendrimer Cross-Linking

Dendrimer cross-linking, resulting in three-dimensional network formation, is one of the simplest approaches for the synthesis of heterogeneous dendrimer-based catalysts, not requiring the special carrier. It can be performed in two different ways. The first is the complex formation with transition metal ions, which are able to bind dendrimer end groups and thus provide the one-step synthesis of the recoverable network dendritic catalysts presenting as coordination polymers [124]. The second way implies the covalent dendrimer cross-linking, using various bi- or polyfunctional agents (diepoxides, diisocyanates, polycarboxylic anhydrides, etc.) [241]. Both of these methods are further considered in the present review.

### 4.1. Coordination Dendritic Network Polymers

In 2009, Maximov and Karakhanov reported the in situ formation of the heterogeneous dendritic coordination network during the Wacker oxidation of linear C_6_–C_12_ alkenes in the presence of CN-terminated PPI dendrimers and 2,2′-biquinoline-4,4’-dicarboxylic acid *N,N′*-bis(3,3′-iminodipropionitrile)diamide (BQC-IPN) (PdSO_4_/CuSO_4_, 80 °C, 5 atm. of O_2_, H_2_O/EtOH, 1 h) [124,241]. It was established that, initially, the homogeneous catalytic system precipitated quickly in the beginning of the reaction, forming agglomerates of 50–200 nm in size, containing small particles of 1.5–2 nm, which were attributed to contracted dendrimer molecules (Figure 29) [124].

It was assumed that Pd^2+^ ions mostly coordinated on the dendrimer terminal C≡N groups, whereas Cu^2+^ ions did so on the internal tertiary amino groups or bipyridyl moieties, resulting in the sterically hindered dendritic network acting as nanoreactors for the selective oxidation of terminal alkenes (Figure 30) [124,241]. An increase in the dendrimer generation and, therefore, in the CN/Pd ratio, led to the decrease in the reaction rate (negative dendritic effect), but favored the selective methylketone formation, reaching 94% and 90% at 1-octene conversions of 37% and 27% for catalysts Pd^II^/Cu^II^/DAB(CN)_16_ and Pd^II^/Cu^II^/BQC-IPN, respectively [241]. Moreover, the use of simple, low-molecular nitrile ligands (CH_3_CN or PhCN) instead of high molecular regular branched dendrimers at the same CN/Pd ratio only retarded the reaction, but did not suppres the oxidation of isomeric alkenes, and inevitably formed in the presence of Pd catalysts [124,241].

Reetz and Giebel developed heterogeneous Lewis acid catalysts based on triflate-terminated PPI dendrimers of the 4th generation, cross-linked with scandium ions (Figure 51) [404]. The synthesized material revealed the high activity and chemo-selectivity in the Mikayama, Diels-Alder, and Friedel-Crafts reactions, giving yields of target product up to 90% (Figure 52), which authors explained in terms of the dendritic matrix swelling in water and, as a consequence, the facilitated substrate access to the inner catalytic centers [404]. The catalyst appeared easily recoverable and maintained its efficacy in several repeated cycles.

Majoral and Caminade suggested the coordination network catalyst, based on phosphazene-cored arylphosphite dendrimers, modified by 15-membered triolefinic triaza-macrocycles on periphery (Figure 53) [405]. Neutral Pd species and clusters originated from Pd_2_(dba)_4_ (where dba was dibenzylidene acetone) or Pd(PPh_3_)_4_ coordinated on C=C double bonds of macrocycles on the dendrimer periphery, thus bonding neighboring dendrimers (Figure 31). Additionally, S=P arising dendrons could form nanoparticle-cored micelles (Figure 31). Hence, there were ensembles obtained typical for other coordination dendrimer-based materials (Figure 32) [124,405].

Depending on the synthesis conditions and certain structure of dendrimer used, both metal complex and nanoparticle (2–5 nm) catalysts were obtained, which were further tested in the Heck cross-coupling reaction between iodobenzene and butyl acrylate (Bu_3_N, THF, 60 °C, 24 h) [405]. In spite of the fact that all catalyst samples were initially isolated and could be recovered as solids, due to the low binding density, most of them behaved as homogeneous catalysts in the process.

The best results with the yield of butyl cinnamate up to 96% were achieved for homogeneous catalysts: metal complex and Pd_7_ cluster. Also, these catalysts appeared easily recoverable and were recycled five times without any loss in activity [405]. This might be due to the well retention of Pd (0) species and small Pd_7_ clusters by the dendrimer-appended macrocycle moieties, and suitable in size. The heterogeneous catalyst was inferior in activity to the homogeneous metal complex and cluster catalysts, giving a yield of butyl cinnamate not exceeding 89%.

A strong negative dendritic effect was revealed: increase in the dendrimer generation from 0th to 4th, as a consequence in the size of Pd particles, resulted in the crucial decrease in conversion and subsequent Pd leaching at catalyst recycling [405]. Pd nanoparticles, located in the inner dendrimer cavities, appeared hardly accessible for substrate due to the multiple steric hindrances arising from the structure of 4th generation dendrimer [405]. Moreover, in the case of aryl phosphite dendrimers, there was possible migration of nanoparticles from the dendrimer inner cavities to the outer surface, and, further, to the reaction medium [406].

### 4.2. Covalent Cross-Linked Dendrimer Networks

Crooks and Bergbreiter, for the first time, used networks based on PPI or PAMAM dendrimers cross-linked with poly(3-methylthiophene) or poly(maleic anhydride)-*alt*-poly(methyl vinylether) copolymer for developing thin films (4–43 nm), appearing to be thermo- and pH-responsive and have electroconductive and selective permeability properties, depending on the treatment conditions [314,407].

Independently, Maximov and Karakhanov suggested the cross-linking of PPI and PAMAM dendrimers with diisocyanates of various size and rigidity, and they were the first who offered to apply the obtained cross-linked dendrimer networks as carriers for heterogeneous nanocatalysts (Figure 54, Figure 33) [241]. Herein, amino, amido, and/or hydroxyl groups in the structure of dendritic network carriers were responsible for the effective binding of metal ions and, as a consequence, for immobilization of metal nanoparticles within a three-dimensional polymeric matrix consisting of the dendrimer moieties in the nodes, bridged with diurethane or aminoethanol ribs. The properties of the resultant materials, such as mean particle size, catalyst activity, and selectivity, were strongly influenced by the dendrimer nature and generation, as well as by the size, rigidity, and polarity of cross-linking agents, and even by the metal deposition conditions [122,127,128,241,287,408,409], thus making dendrimer network carriers partly similar to the metal-organic frameworks (MOF), whose cell parameters and, therefore, the absorption, catalytic, and transport properties were analogously dependent on both size, nature, and rigidity of the organic ribs and in external conditions (temperature, solvent, pressure) simultaneously [410,411].

Thus, PPI and PAMAM dendrimer network-based catalysts were developed, containing Rh, Ru, and Pd nanoparticles successfully applied for hydrogenation of phenols [122,127,412], aromatic [127,408,413,414], and unsaturated compounds under the moderate conditions (50–90 °C, 10–30 atm. of H_2_) [123,128,241,287,409,415]. The main reaction products were the same as for dendrimer-based silica supported catalysts [193,396], and typical for the metal nature under the applied conditions; after hydrogenation of aromatic compounds and phenols in the presence of Rh and Ru containing catalysts, there were corresponding alkylsubstituted cyclohexanes, cyclohexanols, and dihydroxycyclohexanes [1,122,127,408,413,414]. In the presence of Pd-containing catalysts, based on the dendrimer networks, styrenes and dienes were selectively converted to corresponding ethylbenzenes and monoenes with the aromatic ring or conjugated C=C or C=O double bonds remained untouched [123,128,241,287,409,415].

In particular, it was found out that Ru catalysts, based on PPI dendrimers, cross-linked with hexamethylene-1,6-diisocyanate (HMDI) [122], were superior to those of Rh, based on PAMAM dendrimers, analogously cross-linked with 1,6-hexamethylene diisocyanate [412], in the hydrogenation of mono- and dihydric phenols, exhibiting TOF values up to 240,000 h^−1^ (Figure 34) [122]. Analogously, Pd nanoparticles, immobilized in the networks of PPI dendrimers, cross-linked with 4,4′-(3,3′-dimethoxy)diphenyl diisoxyanate (DMDPDI), revealed the efficacy and selectivity in the hydrogenation of phenylacetylene, α,β-unsaturated compounds and conjugated dienes, which were higher than those for Pd catalysts based on DMDPDI cross-linked PAMAM dendrimers (Figure 35) [128,409]. At the same time, PAMAM dendrimer-based catalysts appeared as more active in the hydrogenation of terminal linear alkenes and styrenes (Figure 35) [123,128]. The substrate selectivity for Pd catalysts, based on the networks of both PPI and PAMAM dendrimers, should be remarked, appearing in the highly superior activity for styrenes, dienes, and α,β-unsaturated compounds (i.e., containing the conjugated C=C double bonds) in comparison with linear and cyclic alkenes (1-octene, cyclohexene, etc.) by 15–90 times [128,241].

As for catalysts based on PPI dendrimers and immobilized in the silica pores [396], and those based on the networks of PPI dendrimers exhibited a noticeable competition between steric and electron factors. In most cases, the catalyst activity decreased with the increase in the substrate size and dendrimer generation [122,128,241,408,413] or when passing to the dendrimer network carrier, synthesized with the use of a less expanded linker (e.g., toluene-2,4-diisocyanate (TDI) instead of methylene-bis(*p*-phenylene diiosocyanate) (MPDI)) (Figure 33) [241]. On the other hand, the presence of strong *+I* or *+M*-substituents in the substrate structure [122,408] or the latter closely fitting to the network matrix mesh [409,413] may enhance the catalyst activity even for bulky substrates, such as 4-*tert*-butylphenol [122], 4-*tert*-butylstyrene [413], ethyl- and butylbenzenes, etc. [408]. A positive dendritic effect appearing in the increase in the reaction rate and selectivity with the increase in the dendrimer generation was observed in the hydrogenation of styrenes and phenylacetylene in the presence of Pd catalysts based on PPI dendrimers cross-linked with relatively long and flexible linkers, such as hexamethylene diisocyanate or glycerol triglycidyl ether [287,409], which was possibly due to the better stabilization and higher electron saturation of Pd nanoparticles by multiple donor amino groups in the case of PPI dendrimers of the 3rd generation.

The influence of donor amino groups was found to be essential for the phenol hydrogenation in the presence of Rh and Ru catalysts, especially with regards to hydrogenation of dihydric phenols, such as hydroquinone and resorcinol (Figure 34) [122,412]. In the last case, PPI dendrimers additionally stabilized the enol form for the intermediate product 1,3-cyclohexanedione, making it more liable to the further hydrogenation, thus drastically reducing the negative dendritic effect [122]. Analogously, the higher dendrimer content in the network carriers (40–50% vs. 15–20% for organo-silica carriers with the dendrimers, immobilized in the silica pores [396]) resulted in the stabilization and facilitation of the rotation of the adsorbed enol form of intermediate alkyl-substituted cyclohexanone [396,403], thus favoring a higher portion of the corresponding *trans* hydrogenation product [122]

The use of a polymeric template on the stage of the network carrier synthesis (Figure 55) along with the bulky and rigid dimethoxydiphenyl diisoxyanate linker afforded to develop highly expanded matrix, favoring to the formation of larger particles (5 nm vs. 1.5–2.5 nm, typical for Pd nanoparticles, encapsulated in the networks of PPI dendrimers [287,409]) and, as a consequence, changing their electron properties [128,287]. Thus, there a highly selective catalyst was obtained for phenyl acetylene hydrogenation, maintaining a high styrene yield (~95–96%) even at quantitative conversions, hydrogen pressure of 30 atm., and prolonged reaction times (Figure 36) [128,287]. The maximum TOF values reached up to 185,000 h^−1^ [128,287]; moreover, meso-DAB-PPI-G3-DMDPDI-Pd appeared as the only catalyst, maintaining its activity in the phenylacetylene recycling [287]. Another way to change the catalyst electron properties was applied in the development of bimetallic alloyed PdRu nanoparticles, encapsulated in the network of PPI dendrimers of the 3rd generation, cross-linked with hexamethylene diisocyanate, revealing the enhanced activity exclusively in the benzene hydrogenation (Figure 37) [408].

The influence of templates on the catalyst activity can be observed in the example of Ru catalysts, based on the sterically more hindered PPI dendrimer networks [122], which are significantly inferior to those based on PPI dendrimers, immobilized in the silica pores and thus providing better substrate access to Ru nanoparticles [396]. Moreover, sterically hindered dendritic network carriers appeared to be much more sensitive to the substrate shape and geometry, resulting in better activity for *para*-substituted positively induced phenols, such as *p*-ethylphenol or *p*-*tert*-butylphenol [122,396].

Not only the template, but also the reaction conditions, such as temperature and solvent, on the stage of nanoparticle encapsulation had a great influence on the physical chemical properties and, as a consequence, on the hydrogenation activity, which was demonstrated on the example of Ru catalysts, based on PPI dendrimers cross-linked with poly(ethylene glycol) (PEG) diglycidyl ether [127]. Consisting of the blocks with differing low critical solution temperatures (LCST) (PEG chains and PPI dendrimers), these catalysts revealed different thermos-responsive properties on the hydrogenation of both phenol and benzene, essentially depending on the reaction conditions on the stages of both catalyst synthesis and subsequent hydrogenation. It was found out that the use of the “bad” reaction conditions in the catalyst synthesis (inappropriate, “bad” solvent, e.g., THF instead of water, or temperature above LCST for polymer blocks, constituting the network support) favored the superior activity in the hydrogenation of phenol and, especially, of benzene, but resulted in the partial or complete loss of thermo-responsive properties [127] similar to thin films, based on PPI or PAMAM dendrimers, cross-linked with poly(maleic anhydride) [314].

Deposition of Rh or Pd nanoparticles on the networks of PPI and PAMAM dendrimers under the conditions of supercritical CO_2_ (scCO_2_) resulted in the extremely efficient catalysts, exhibiting TOF values up to 280 s^−1^ (>1,000,000 h^−1^) and 207 s^−1^ (>745,000 h^−1^), respectively, at the metal loadings as low as 0.01–0.29% [416]. Herein, Rh catalysts, based on PAMAM dendrimers, revealed the highest activity in the hydrogenation of alkenes, such as 1-octene and, especially, styrene, and a noticeable negative dendritic effect was observed for catalysts on both PAMAM and PPI dendrimers. Pd catalysts appeared to be the most effective in the hydrogenation of conjugated dienes [415].

The presence of water was found to be crucial for hydrogenation of phenols and aromatic compounds in the presence of Ru catalysts [122,408,413]. Simultaneously, the dendritic matrix, cross-linked with diisocyanates, was subjected to Ru-catalyzed hydrolysis [122]. Nonetheless, the catalysts, based on the dendritic networks, can be successfully reused several times without a significant loss in activity and selectivity under both aqueous (Ru, Rh) and non-aqueous conditions (Pd) [122,128,287].

Moreover, the network matrices, based on PPI dendrimers, appeared to be a suitable carrier for Ni-W sulfide catalysts, and successfully applied for the hydrogenation of naphthalene (350–400 °C, 50 atm. of H_2_, 5 h) [414]. The catalyst efficiency was found to be dependent on the synthesis conditions (ex situ or in situ), process temperature, and sulfidizing agent, and the best results (55% selectivity on decalins and 45% selectivity on tetralin at 98% conversion of naphthalene) were reached at 380 °C using the catalyst synthesized in situ and elemental sulfur as the sulfidizing agent [414]. It should be noted that PPI dendrimers may undergo retro-Michael addition under hydrogenation conditions even at temperatures below 100 °C [61]; as a consequence, dendrimer network carriers, at least, were partially decomposed at 350–400 °C and 50 atm. of H_2_, under which the process was performed [394]. This might be an additional reason as to why in situ catalysts appeared more efficient.

Ogasawara and Kato developed a mesoporous polymer Tailor-made catalyst based on PAMAM dendrimer-encapsulated nanoparticles [417]. This approach suggested the formation of small dendrimer-encapsulated Pd nanoparticles (~2 nm*)* in situ, simultaneously with the synthesis of the cross-linked polymeric matrix, outside the cavities, formed by PAMAM dendrimers (Figure 56). The catalyst synthesized revealed high activity in the Suzuki cross-coupling between phenylboronic acid and 4-bromoacetophenone (80 °C, K_2_CO_3_, H_2_O), giving a quantitative conversion within 4 h with TOF values up to 2500 h^−1^ [417]. Replacement of the *–M*-acceptor acetyl group by the C≡N group, possessing with the strong coordination effect, or by +*M*-donor OH or OCH_3_ groups, resulted in a noticeable decrease in the catalyst activity. The catalyst exhibited good resistance to metal leaching and can be recycled eight times, maintaining the yield of cross-coupling target product of more than 90% [417].

The similar approach was applied by Nabavinia, Kanjilal, and Noshadi for the synthesis of PEI dendrimer-encapsulated palladium nanoparticles, encaged into the resorcinol formaldehyde micro/mesoporous resin polymer network (MPR) in situ in the absence of the polymer template (Figure 57) [416]. The catalyst activity and recyclability in the Suzuki cross-coupling reaction between phenylboronic acid and toluilhalide was found to be strongly dependent on the conditions of both catalyst synthesis (the solvent used) and cross-coupling reaction (arylhalide nature, temperature, solvent). The maximum efficacy was observed for the catalyst synthesized in water-ethanol medium, and the quantitative conversions for 4-iodotoluene were observed already within 30–50 min at 85 °C using water-ethanol medium again and K_2_CO_3_ as a base. For 4-bromotoluene, the conversion of about 80% was reached within 5–6 h at 110 °C using water-DMF medium and K_2_CO_3_ as a base. The synthesized PEI-MPR-Pd catalysts revealed high thermal stability and good resistance to metal leaching and thus can be recycled several times without a noticeable loss in activity and used in the continuous-flow micro set-up, with conversion reaching 62% at 85 °C and a flow rate of 0.2 μL/min [416].

## 5. Magnetically Separable Heterogeneous Dendrimer-Based Catalysts

The use of nanoparticles and carriers possessing ferromagnetic properties, opens the way for versatile design of the easily and effectively recoverable catalysts, essentially diminishing the metal losses and, therefore, the cost of the process and target products [10,62,64,418,419]. Moreover, the presence of iron or nickel oxide in the nanoparticle core provides not only the easy catalyst separation, but also is able to enhance the catalytic activity selectivity, and even to change the reaction pathway [418] that has been demonstrated on the examples of Ru-catalyzed nitrobenzene hydrogenation [420], Zn-catalyzed methanol synthesis from syngas [421], selective Ru-catalyzed hydrogenolysis of cellulose to ethylene glycol in water [422], and zeolite ZSM-5 catalyzed selective methanol conversion to C_5_–C_8_ and C_9_–C_11_ hydrocarbons (MTH) [423].

With respect to dendrimer-based materials, magnetic separation can be applied to both homogeneous and heterogeneous catalysts [10,62,64,419]. In the first case, it is realized through the dendrimer-encapsulated mono- or bimetallic core-shell nanoparticles, where the core is magnetically active metal or metal oxide (Fe, Co, Ni) and the shell is a catalytically active metal (Ru, Pt, Pd, Rh etc.) (Figure 38, left); on the other side, there can be a synthesized nanoparticle-cored dendrimer, where there is magnetically active metal or metal oxide again, while catalytic centeres are located elsewhere in the dendron moiety (Figure 38, right) [419,424,425]. Thus, the iron oxide (Fe_2_O_3_) nanoparticle cored melamine dendrimer, functionalized with Pd phosphine complexes on the termini (Figure 39), proved its efficacy in the Suzuki cross-coupling reaction between various arylhalide and *meta*-substituted arylboronic acids (K_2_CO_3_, CH_2_Cl_2_/DMF, 12 h), giving yields of 72–81%, depending on the halide and substituent nature [425]. In spite of the occurrence as homogeneous catalyst, this hybrid material appeared easily recoverable under the external magnetic field and was used in the subsequent reaction cycles without any loss in activity [425].

Heterogeneous dendrimer-based magnetically separable catalysts can be synthesized in different ways [419]. Nonetheless, in the structure of such catalysts there can be distinguished the following main constituents (Figure 39) [419,425]: (1) the magnetic core (Fe_2_O_3_, Fe_3_O_4_, Co, Ni), which may be additionally coated with silica or polymer; (2) dendrimers or dendrons, immobilized on the magnetic core using covalent attachment (convergent or divergent), π–π interactions, or hydrogen bonds; (3) catalytic centers (metal complexes or nanoparticles), located inside or outside the dendrimer moiety. Several examples for the synthesis and application of magnetically separable dendritic catalysts are further considered in the present review.

### 5.1. Ferrous Oxide Core Magnetically Separable Dendritic Catalysts

Bronstein et al. developed hybrid magnetically sensitive material Den-Fe_3_O_4_ based on Müllen-type polyphenylene pyridyl-terminanated dendrons of the 1st generation, covalently attached to iron oxide nanoparticles of 18–25 nm in size (Figure 58) [426,427,428]. Subsequent impregnation with Pd^2+^ ions, coordination on pyridyl moieties, followed by molecular hydrogen reduction, resulted in well dispersed Pd nanoparticles of 0.9–1.9 nm in diameter, uniformly distributed on the surface of iron oxide (Figure 40) [426,427,428,429]. Herein Pd species functioned as peculiar linking nods between dendrimer-coated iron oxide nanoparticles—similar to Pd^2+^, Cu^2+^ and Sc^3+^ ions in the dendrimer coordination network catalysts [124,241,404]—thus facilitating the catalyst aggregation and separation under the magnetic field [426].

The synthesized catalyst Pd-Den-Fe_3_O_4_ proved its efficacy in the Suzuki cross-coupling reaction between phenylboronic acid and various 4-bromoarenes [426,427] and semi-hydrogenation of dimethylethynylcarbinol (DMEC) to dimethylvinylcarbinol (DMVC) [428,429]. In the last case, the yields of target product and TOF values as high as 90–98% and 5760–33,480 h^−1^, respectively, were reached already after 0.5–1 h at 90 °C and atmospheric hydrogen pressure (Figure 59a) [428,429]. Moreover, tha Pd-Den-Fe_3_O_4_ catalyst maintained its activity and highest selectivity on dimethylvinylcarbinol in several repeated cycles [428,429].

The best yields as high as 90% in the Suzuki cross-coupling reaction between phenylboronic acid and 4-bromoanisole were reached after 1 h at 70 °C, using K_2_CO_3_ as a base and dioxane/water mixture as a reaction medium [426]. The use of hyperbranched pyridylphenylene polymer (PPP), cross-linked with diphenyl ether moieties and Pd(OAc)_2_, supported on magnetic silica (MS), allowed it to reach quantitative yields in Suzuki cross-coupling with electron withdrawing substrates (4-bromobenzaldehyde, 4-bromonitrobenzene) already within 3–15 min with TOF values up to 23,500 h^−1^ (H_2_O/EtOH, Na_2_CO_3_, 60 °C) [427].

Moreover, the catalyst Pd-Den-Fe_3_O_4_ appeared suitable for the process proceeding in the continuous-flow microreactor where catalyst beds were immobilized, using the constant magnetic field (Figure 59b) [426]. The best results (conversion of 17–18% and selectivity of 53–55% on 4-phenylanisole) were achieved at 70 °C, flow rate of 0.2 mL/min, and pressure of 5 atm. for double-sided glass reactor and at 90 °C, flow rate of 0.2 mL/min, and pressure of 10 atm. for double-sided stainless-steel reactor, respectively. Herein, no Pd leaching was observed in the eluate after storing the samples for 1 week [426].

Rosario-Amorin, Nlate, and Heuzé suggested the Pd complex catalysts based on bis(aminomethyl-*P,P*-dialkylphosphine) terminated poly(alkyl aryl ether) dendrons covalently grafted to γ-Fe_2_O_3_ nanoparticles of 300 nm in diameter, coated with polycarboxylic acid (Figure 41) [430]. These catalysts revealed a noticeable positive dendritic effect in the Suzuki cross-coupling reaction between arylbromides and phenylboronic acid at 65 °C, using NaOH a base and THF/triton X405/water as a reaction medium (Figure 60). The maximum yields, as high as 100% within 0.5–1 h, were achieved for arylbromides with strong −*M*-*para*-substituents (-C(=O)H, -NO_2_), whereas +I-substituents, especially in ortho-position to Br or B(OH)_2_ group in the substrate, vice versa, retarded the reaction. Nonetheless, all of the synthesized hydride dendritic catalysts appeared easily recoverable by means of magnetic separation and successfully reused in 25 reaction cycles with just a slight decrease in conversion (100% → 95%) [430].

Astruc and co-workers have designed well-recoverable materials based on PEG-terminated poly(carbosilane/triazol/aryl ether) dendrimers and dendrons, both covalently attached (Figure 42) and immobilized by means of hydrogen bonds on the surface of silica coating Fe_3_O_4_ or Fe_2_O_3_ nanoparticles (Figure 61), called superparamagnetic iron oxide nanoparticles (SPIONs) [10,64,431,432,433]. Impregnated with Pd nanoparticles with a mean size of 1.5–3 nm, these catalysts appeared very effective in the various cross-coupling and alcohol oxidation reactions under moderate conditions (Figure 62), giving the quantitative yields after 24 h at 60–100 °C and TOF values up to 920 h^−1^ [432,433]. A strong positive dendritic effect on the catalyst activity and stability was revealed, making possible a successful recovery under the magnetic field and subsequent repeated use up to 10 times [432,433], thus significantly surpassing nondendritic magnetically separable catalysts with γ-Fe_2_O_3_@SiO_2_ core [431]. Herein, bulk dendritic fragments afforded the localization of Pd nanoparticles near the SPION core surface, thus making the catalyst more robust and stable [431].

Alper at al. suggested well recoverable Rh-containing catalysts, based on PAMAM dendrons covalently attached to silica coated iron oxide Fe_3_O_4_ (Figure 43) using a divergent strategy for styrene hydroformylation [419,434]. The quantitative conversions and branched/linear ratios of more than 40:1 after 16–20 h were reached at 40–50 °C, 70 atm. of CO/H_2_ (1:1), using CH_2_Cl_2_ as a reaction medium, and the highest branched product yield of up to 99% were obtained for the substrates with −*M* substituents (e.g., 4-vinylbenzoic acid). An increase in the dendron generation influenced slightly negatively on the branched/linear ratio (44:1 for G1 and 40:1 for G3), but rather positively on the catalyst stability, and conversions of 98–100% maintained during 5 reaction cycles [434].

Os (VI) heterogenized metal complex catalysts, based on the tetraalkylammonium salts, covalently attached to silica coated iron oxide Fe_3_O_4_ via (3-aminopropyl)tetraoxysilane bridge and additionally stabilized by poly(aryl ether) dendrons, where the latter one of the substituents at quaternary nitrogens (Figure 44) were successfully applied in the dihydroxylation of both linear and cyclic olefins (Figure 63) [435]. The diol yields of 87–97% were achieved after 2–9 h under mild conditions (methylmorpholine *N*-oxide (NMO) as an oxidant, room temperature, acetone/water as a reaction medium), thus being significantly superior to the control catalyst, giving a yield of 85% after 5 h (Figure 44). The positive dendritic effect on both the reaction rate and catalyst stability was found, and the catalyst, based on the dendron of the 2nd generation, proves its reusability in the oxidation of various alkenes, maintaining the diol yields of 90–97% during 5 cycles at least [435].

### 5.2. Co Core Magnetically Separable Dendritic Catalysts

Caminade, Majoral, and Ouali developed magnetically separable Pd complex catalysts, formed in situ by the interaction of palladium (II) acetate with diphenylphosphite-terminated polyphosphazene dendrimers [436]. The catalyst heterogenization was accomplished using pyrene moiety at the dendron focal point via π–π interactions with graphene-coated cobalt magnetic nanoparticle (Co MNP) (Figure 64) [419]. Thus, synthesized catalysts were successfully applied in the Suzuki cross-coupling reaction (Figure 65). The quantitative conversions were reached for −*M* (-C(=O)CH_3_, (-C(=O)OH) *para*-substituted arylbromide, coupled with unsubstituted phenylboronic acid, after 15 h at 60 °C, using Na_2_CO_3_ as a base, H_2_O/THF mixture as a reaction medium, and zero generation dendrons as a stabilizing ligand [436]. During the reaction, the catalyst immobilized and turned to a homogeneous form and was easily recovered under the external magnetic field by the temperature cooling down to 20–25 °C, thus resulting in the π–π interactions renewal [419,436]. Hence, using this release and catch strategy, the catalyst can be successfully reused up to 12 times without any loss in activity [419,436].

## 6. Conclusions

Dendrimers are a unique class of organic compounds. Due to their branched, regular structure, and a variety of types and possibilities for modification, dendrimers have found wide applications in nanomedicine, chemical engineering, light and electron sensing devices, and, especially, in nanocatalysis. The present review has compiled the common methods for the synthesis and application of the dendritic catalysts, especially emphasizing the advances in design of the heterogeneous dendrimer-based materials during the last two decades. As one can overview, in spite of the variety of the synthetic routes, developed for heterogeneous dendritic catalyst, most of them can be narrowed down to three main approaches.

The simplest is the wetness impregnation of various heterogeneous carriers, such as ordered mesoporous silica, carbon black, carbon nanotubes, or graphite with the colloidal solution of dendrimer-encapsulated nanoparticles. The resistance to metal leaching and, therefore, the catalyst recyclability are provided, and on the one hand, by hydrogen bonds or π–π interactions between dendrimer molecules and carrier, and, on the other hand, by the retention of metal nanoparticles inside the cavities of higher generation dendrimers—similar to homogeneous dendritic catalysts. Thus, synthesized catalysts appeared suitable for the severe industrial conditions and processes, such as hydrocarbon hydrogenation, dehydrogenation, ring-opening, isomerization, oxidation, etc.

The second approach is the covalent grafting of dendrimers or dendrons to heterogeneous carriers (amorphous and mesoporous silica, polystyrene, carbon nanotubes), thus stabilizing the dendritic ligand. This strategy can be realized both in convergent and divergent ways; the convergent way, using ready-prepared dendrimers and dendrons, seems to be advantageous, enhancing the coating yield and diminishing the defects in the structure of future material. Herein, the certain synthetic route strongly depends on both the dendrimer and carrier nature. Dendrimer anchoring to silica-based supports results in the hybrid organic-inorganic, or organo-silica materials.

The third approach implies the synthesis of dendrimer networks carriers. Dendrimers can be bound here by transition metal ions, such as Pd, Cu, Sc, etc., forming coordination polymer networks, acting as acid-bases and metal complex catalysts. Behavior of such catalysts is strongly dependent on the dendrimer nature. Thus, sterically hindered PPI dendrimer of higher generations (3rd and 4th) are able to provide the best reaction yield and selectivity. Vice versa, phosphazene-cored aryl phosphite dendrimers are much more effective at zeroth generation. Otherwise, dendrimers can be cross-linked with diepoxides, diisocyanates, etc., thus resulting in fully organic heterogeneous carriers, which can be speculated as a peculiar analogue of metal-organic frameworks. The properties of these materials and their behavior in catalysis are strongly dependent on the dendrimer nature and generation, on the one side, and on linker size, rigidity, and polarity, on the other side.

Finally, magnetically separable materials can be synthesized, using both weak interactions (hydrogen bonds, π–π interactions) and covalent grafting between dendrimer or dendron and magnetic cores, often coated with silica, carbon, or polymer shell.

Metal appending for both grafted dendritic materials and dendrimer networks is conducted similar to procedures for the synthesis of homogeneous dendrimer-based catalysts, and complex formation between the dendrimer functional node and/or end groups and metal ions or low-molecular complexes is the driving force for the metal impregnation inside the dendritic carrier. Both monometallic and bi- or polymetallic heterogeneous dendritic catalysts can be synthesized in this way. Among others, polyamine dendrimers (PAMAM, PPI, PETIM, PEI) are characterized by commercial ability, relatively simple synthetic procedure and, first of all, the large amount of donor groups able to catch metal and complex formation; hence, the overwhelming majority of the catalyst examples, illustrated in the review, refers to dendrimers of the said types.

Heterogenized dendrimer-based catalysts are obtained and stored as solids/powders similar to conventional heterogeneous catalysts. Nonetheless, various examples in hydrogenation, hydroformylation, oxidation, cross-coupling reactions, etc. have revealed them not only to maintain their efficacy at recycling, being easily recoverable, but also to exhibit performance, often superior to analogous homogeneous catalysts, with TOF values reaching up to 400,000 h^−1^. Herein, dendrimer moieties acted as nanoreactors, preventing nanoparticle agglomeration and metal leaching and providing a high reaction rate and selectivity. Reaction in these nanoreactors may proceed in a pseudo-homogeneous way on small nanoparticles and clusters (nanoheterogeneous catalysis) or on the metal complex centers, attached to heterogenized dendrimers or dendrons. The “Release and catch mechanism, suggesting the temporary process transition to homogeneous phase, can also take place. Moreover, polyamine dendrimers (PPI, PAMAM, PETIM) in water medium may act as acid-base catalysts, thus additionally increasing the reaction rate and directing the reaction pathway.

Depending on both dendrimer and reaction type, as well as on the substrate electronic properties, together influencing the stability and accessibility of catalytically active species, a positive or negative dendritic effect can take place. In some cases, e.g., semi-hydrogenation of alkynes or conjugated dienes or hydrogenation of phenols, the selectivity can be affected by both the dendrimer type—similar to homogeneous dendrimer-based catalysts—and the dendrimer portion in the carrier.

The catalysts, based on the dendrimer networks, should be especially remarked, as their properties (mean particle size and distribution, thermoresponsivity, efficacy and selectivity) have appeared to be directly influenced by several factors simultaneously. Those are the dendrimer type and generation; linker size, rigidity, and polarity; reaction conditions for the network synthesis and metal deposition (temperature below or above LCST, “good” or “bad” solvent, the presence or absence of polymeric template, supercritical medium etc.). Varying these parameters, one can obtain the efficient catalysts for hydrogenation of linear alkenes to alkanes or, vice versa, of phenylacetylene to styrene—even at quantitative substrate conversions.

Hence, heterogeneous dendrimer-based catalysts have proved their efficacy and recyclability in various chemical processes. In spite of many advances in this field, the design of heterogeneous dendritic catalysts is still developing, featuring versatile approaches.

## Data Availability

Not applicable.

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
