# Peer review of "Heterogeneous Dendrimer-Based Catalysts"

_polymers, 2022, doi:10.3390/polym14050981_

Round 1

Reviewer 1 Report

This manuscript deals with a dendrimer-catalyst composite materials.  The authors discussed "What is dendrimer?", "How to use dendrimer?" Also, they classified the dendrimer-catalyst composite materials with an enough reports.  Therefore, the reviewer thinks this review is published in the Journal.

However, there are several mistypes in the manuscript: L25, anines, phoshines, and thyols, and L134, aminoogroups.  Since the manuscript is very long, the reviewer could not check it thoroughly.  The reviewer suggests that  this manuscript would be proofheaded.

Author Response

Thank you for your revision.

The mistakes revealed have been corrected.

Reviewer 2 Report

The manuscript by Zolotukhina et al. provides a comprehensive review of the various ways in which dendrimers have been involved in heterogeneous or supported catalysis. The manuscript has been carefully prepared and, to the best of my knowledge, covers the essential aspects and references of the field. The introduction and conclusions are useful for the reader who wants to know the state of the art. The organization of the body of the paper by type of immobilization is coherent and facilitates reading. However, it has the disadvantage of dispersing information on applications in specific catalytic processes. This is my main criticism: considering the organization of the manuscript, the information given about catalytic results is in some cases overdetailed and may disorient rather than help the reader. In any case, this does not change my positive opinion about this review. As a minor point, I only suggest that the authors modify the end of the introduction to make it clearer that points 1 to 3 reflect the organization of the review, for instance, adding in parentheses the number of the section corresponding to each epigraph. In addition, the authors should revise the numbering of the sections, since there are two sections numbered 3.

Author Response

The manuscript by Zolotukhina et al. provides a comprehensive review of the various ways in which dendrimers have been involved in heterogeneous or supported catalysis. The manuscript has been carefully prepared and, to the best of my knowledge, covers the essential aspects and references of the field. The introduction and conclusions are useful for the reader who wants to know the state of the art. The organization of the body of the paper by type of immobilization is coherent and facilitates reading.

Answer: Thank you for your kind revision.

However, it has the disadvantage of dispersing information on applications in specific catalytic processes. This is my main criticism: considering the organization of the manuscript, the information given about catalytic results is in some cases overdetailed and may disorient rather than help the reader. In any case, this does not change my positive opinion about this review.

Answer: It is just my opinion. As we talk about catalysts, we cannot omit their efficacy and application. If this review had been devoted to dendrimer-based materials in whole (there are many such reviews), I could just briefly mention their application due to its large diversity.

As a minor point, I only suggest that the authors modify the end of the introduction to make it clearer that points 1 to 3 reflect the organization of the review, for instance, adding in parentheses the number of the section corresponding to each epigraph.

Answer: The end of the Introduction has been modified in the following way:

“By the present time three main approaches for the synthesis of heterogeneous dendrimer-based catalysts have been developed and are further considered in the present review.

The first is the adsorption of metal-containing dendrimers or dendrons on the surface of amorphous silica, carbon or inside channels in channels of mesoporous materials via hydrophobic, electrostatic or π–π interactions [86]. This approach is considered in the Section 2.

The second approach supposes the covalent grafting of dendrimers or dendrons on polymers, amorphous silica, carbon nanotubes or inner walls of mesoporous materials [86]. It is considered in the Section 3.

The third approach, including the dendrimer cross-linking, using various bi- or trifunctional agents, polymers etc. [242] is discussed in the Section 4.

Additionally, magnetically–separable dendritic catalysts [62], which may be designed, using all three approaches mentioned above, are considered in the Section 5”

In addition, the authors should revise the numbering of the sections, since there are two sections numbered 3.

Answer: The numbers of sections have been corrected.

Reviewer 3 Report

This paper is a very large review (429 references) which contains a lot of useful information about “heterogenous dendrimer-based catalysts”. Undoubtedly, this review has necessitated a huge work to be written, and is worth to be published, but a lot of improvements are needed.

I have a few remarks, the most important concerning the abstract:

  • only 3 lines of abstract for 106 pages of text, it is certainly a record! The abstract should be amplified, explaining what can be found in the review. It is not presently the case. Indeed, the word “heterogeneous” can concern different topics, the simplest being a catalytic dendrimer used in a solvent in which it is not soluble, but this aspect is not treated in this review. It is difficult to give more explanations in the title to explain the meaning of “heterogenous” thus the abstract should give more information.
  • Normally, when citing a name from a reference, it is either the first author, or the stared author(s). Thus, the arylphosphite dendrimers should be indicated as Caminade/Majoral type.
  • The paragraph after Figure 6 concerns “nanoparticle and metal complex dendrimer-based catalysts”, and contains probably more than 200 references. However, a lot of references (several hundreds) are missing concerning “metal complex dendrimer-based catalysts”. Thus, I suggest to concentrate in this paragraph about “nanoparticle dendrimer-based catalysts”, as all the following pages concern nanoparticles either as catalysts or as supports of dendrimers or dendrons. It will be easier for the reader, and also is could be possible to give a full overview about nanoparticle dendrimer-based catalysts. A few references are missing, for instance: Mater. 1999, 11 (3), 217-220; Angew. Chem. Int. Ed. 1999, 38 (3), 364-366; Langmuir 2008, 24 (5), 2090-2101; Proc. Natl. Acad. Sci. U. S. A. 2012, 109 (29), 11493-11497; Surf. Sci. 2015, 640, 65-72; and certainly others.
  • It is written that “homogeneous dendrimer-based catalysts can be stored and used mostly as colloidal solutions, undergoing gradual deactivation due to metal leaching and precipitation from the dendrimer stabilizing micelles”. It is true in some cases, but not always. On the contrary, it has been shown that dendrimer complexes induce a considerable decrease of the leaching (see in particular Green Chem. 2012, 14 (10), 2807-2815).
  • The caption to Scheme 6 is misleading: “dendrimer-encapsulated nanoparticles onto the surface of mesoporous carrier” should be replaced by something like “dendrimer-encapsulated nanoparticles inside a mesoporous carrier”
  • In the work described lines 1645-1651, I suppose that at 380°C, the dendrimer no longer exists. This should be indicated.

The paper is well-illustrated with many Schemes and Figures, but I have a few comments about some of them:

  • In Figure 1, one can imagine that all dendrimers, except the phosphazene dendrimer, and the phenylazomethine dendrimer, are synthesized by divergent processes, because they have potentially reactive terminal functions. But reality is different. The poly(aryl ether) dendrimer is synthesized by a convergent way, and the phosphazene dendrimer by a divergent way. Thus, instead of presenting the poly(aryl ether) dendrimer, a poly(aryl ether) dendron should be presented, with phenyl groups (no OH) as terminal functions. On the other side, an aldehyde should be added in para position on all the terminal aryl groups of the phosphazene dendrimer.
  • In Figure 5, there are many overlaps between oxygen atoms, it is not very beautiful.
  • In Schemes 9, 10, and 39, the drawings represent dendrons attached to the surface of nanoparticles, but the captions to these Schemes indicate that these are dendrimer-encapsulated nanoparticles. I suppose that the captions are right, and that the schemes should be modified. In the other Figures and Schemes, the difference between dendrons and dendrimer is clear, so it should be the same for these 3 Schemes.
  • In Figure 19, the real chemical structure of the polyaromatic framework is unclear. Why not drawing also the chemical structure?
  • In Figure 21, the chemical structure of the dendrons is not given. It should be given.

The paper is well organized, but the writing should be improved. There are many sentences which are exceedingly long (more than five lines, in some cases 10 lines), thus they should be split in at least two sentences. There are also a few systematic misprints:

  • “Phenylazomethyne” should be replaced by “phenylazomethine” (6 occurrences)
  • “Aminogroup” (or aminogroups) should be replaced by “amino group” (or amino groups) (12 occurrences)
  • Delete “to” in “favoured to” (8 occurrences)

And many other misprints:

  • Lines 23-24: in the sentence “To prevent metal nanoparticle sintering and agglomeration, add, as a consequence, catalyst deactivation during the process, various organic ligands, such as ionic liquids”, “add” should be replaced by “and”
  • Line 25: “thyols” should be replaced by “thiols”
  • Line 167: replace “poly(ethyele imine)” by “poly(ethylene imine)”
  • Line 273: replace “subnanoclasters” by “subnanoclusters”
  • Line 386: what is “Рукуштcarbon”?
  • Line 571: replace “phospine“ by “phosphine”
  • The sentence line 590 “SiO2-PAMAM-Rh catalysts appeared slight negative dendritic effect” should be modified such as “SiO2-PAMAM-Rh catalysts display a slightly negative dendritic effect”
  • Lines 621-622: in “under to carbonylation”, delete “to”
  • Line 625: replace “bromorenes“ by “bromoarenes“
  • Line 630: replace “The same type catalysts” by “The same type of catalysts”
  • Lines 633-634: “The maximum yields of 85–99% were observed for the substrates with electron withdrawing groups (C(=O)OMe, NO2), maintained at catalyst recycling” has to be modified, for instance as “The maximum yields of 85–99% were observed for the substrates with electron withdrawing groups (C(=O)OMe, NO2), and were maintained when recycling the catalyst”
  • Line 745: in “conventional thermal impact“, replace “impact” by “heating”
  • Line 771: replace “иby” by “by”
  • Line 816: replace “the use of sterically hindrance ortho-substituted” by “the use of sterically hindered ortho-substituted”
  • Line 883: replace “in differing ways” by “in different ways”
  • Line 1054: replace “propogation” by “propagation”
  • Line 1064: replace “80 0C” by “80°C”
  • Line 1171: replace “ompregnation” by “impregnation”
  • Lines 1307: in the sentence “Replace of 1-octyne”, “Replace” should become “Replacement”
  • Line 1590: replace “PPI dendrimera” by “PPI dendrimer”
  • Line 1622: replace “revealing” by “revealed”
  • Line 1628: in “but resulted in the in the partial” delete one “in the”
  • Line 1788: replace “dendonized” by “dendronized”
  • Lines 1854-1855: replace “diphenylphoshite-terminated” by “diphenylphosphite-terminated”
  • Line 1900: in “The third approach implies is the synthesis”, choose between “implies” or “is”, but don’t use both.

And probably many other misprints I missed. This paper should be very carefully re-read by the authors.

Author Response

This paper is a very large review (429 references) which contains a lot of useful information about “heterogenous dendrimer-based catalysts”. Undoubtedly, this review has necessitated a huge work to be written, and is worth to be published, but a lot of improvements are needed.

Answer: Thank you for your kind revision and careful reading.

I have a few remarks, the most important concerning the abstract:

    only 3 lines of abstract for 106 pages of text, it is certainly a record! The abstract should be amplified, explaining what can be found in the review. It is not presently the case.

Answer: The abstract has been amplified as follows:

“The present review comprehends the advances in the dendritic catalysis within last two decades, in particular concerning heterogeneous dendrimer-based catalysts and their and application in various processes, such as hydrogenation, oxidation, cross-coupling reactions etc. There are considered three main approaches to the synthesis of immobilized heterogeneous dendrimer-based catalysts: 1) impregnation/adsorption on silica or carbon carriers; 2) dendrimer covalent grafting to various supports (silica, polystyrene, carbon nanotubes, porous aromatic frameworks etc.), which may be performed in a divergent (as a gradual dendron growth on the support) or convergent way (as a grafting of whole dendrimer to the support); and 3) dendrimer cross-linking, using transition metal ions (resulting in coordination polymer networks) or bifunctional organic linkers, whose size, polarity and rigidity define the properties of the resulted material. Additionally magnetically separable dendritic catalysts, which can be synthesized using three above mentioned approaches, are also considered. Dendritic catalysts, synthesized in such ways, can be stored as powders and be easily separated from the reaction mixture by filtration/centrifugation as traditional heterogeneous catalysts, maintaining efficiency as for homogeneous dendritic catalysts.”

Indeed, the word “heterogeneous” can concern different topics, the simplest being a catalytic dendrimer used in a solvent in which it is not soluble, but this aspect is not treated in this review. It is difficult to give more explanations in the title to explain the meaning of “heterogenous” thus the abstract should give more information.

Answer: Yes, sometimes typical homogeneous dendritic catalyst may behave as heterogeneous (e.g. in the “bad” solvent), like in the work Nano Lett. 2003, 3, 1757-1760. Sometimes the solvent adjusted by affinity to substrate/product, not to catalyst ligand.

As a rule, “bad” solvent is used to recover homogeneous dendritic catalysts from the solution for subsequent repeated use. More frequently homogeneous dendritic catalysts are used under two-phase conditions or in the solvent, “good” for the substrate and dendrimer simultaneously, not in the neat substrate.

These speculations have been added in the Introduction:

“… As a consequence, since the end of 1990s various approaches for heterogenization of dendrimer-based catalysts have been suggested. These heterogenized dendritic catalysts catalysts are supposed to exist as powders, easily precipitate and isolate from the reaction medium using filtration or centrifugation – similar to conventional heterogeneous catalysts. In some cases homogeneous micellar dendritic catalysts, initially obtained as solids, may be used as heterogeneous in the “bad” solvent medium [121]. Immobilization of dendritic catalysts, using various heterogeneous carriers, such as silica, polystyrene etc. allows to significantly enhance their stability and recyclability. 

By the present time three main approaches to the synthesis of immobilized heterogeneous dendrimer-based catalysts have been developed and are further considered in the present review…”

    Normally, when citing a name from a reference, it is either the first author, or the stared author(s). Thus, the arylphosphite dendrimers should be indicated as Caminade/Majoral type.

Answer: Thank you for this remark. The known stared authors are added to dendrimers of corresponding types in the corrected Figure 1.

    The paragraph after Figure 6 concerns “nanoparticle and metal complex dendrimer-based catalysts”, and contains probably more than 200 references. However, a lot of references (several hundreds) are missing concerning “metal complex dendrimer-based catalysts”. Thus, I suggest to concentrate in this paragraph about “nanoparticle dendrimer-based catalysts”, as all the following pages concern nanoparticles either as catalysts or as supports of dendrimers or dendrons. It will be easier for the reader, and also is could be possible to give a full overview about nanoparticle dendrimer-based catalysts. A few references are missing, for instance: Mater. 1999, 11 (3), 217-220; Angew. Chem. Int. Ed. 1999, 38 (3), 364-366; Langmuir 2008, 24 (5), 2090-2101; Proc. Natl. Acad. Sci. U. S. A. 2012, 109 (29), 11493-11497; Surf. Sci. 2015, 640, 65-72; and certainly others.

Answer: The said works have been mentioned in the revised manuscript and added to reference list. The work Langmuir 2008, 24 (5), 2090-2101 has been considered more thoroughly in the section, devoted to coordination dendrimer networks, because we suppose, that Pd (0) species here may bind neighboring dendrimers. Also a table, distinguishing the examples of metal complex and nanoparticle dendritic catalysts, is added to the Introduction.

    It is written that “homogeneous dendrimer-based catalysts can be stored and used mostly as colloidal solutions, undergoing gradual deactivation due to metal leaching and precipitation from the dendrimer stabilizing micelles”. It is true in some cases, but not always. On the contrary, it has been shown that dendrimer complexes induce a considerable decrease of the leaching (see in particular Green Chem. 2012, 14 (10), 2807-2815).

Answer: Yes, this is an interesting work. But as far as we have understood from the Experimental procedure description, this catalysts is prepared in situ and then, after reaction, is recovered by adding the “bad solvent”, that is typical for homogeneous dendrimer-based catalysts. Nonetheless, this work has been mentioned in the revised manuscript and added to reference list.

    The caption to Scheme 6 is misleading: “dendrimer-encapsulated nanoparticles onto the surface of mesoporous carrier” should be replaced by something like “dendrimer-encapsulated nanoparticles inside a mesoporous carrier”

Answer: Done.

    In the work described lines 1645-1651, I suppose that at 380°C, the dendrimer no longer exists. This should be indicated.

Answer: This is the quote from the article “Dendrimer-Encapsulated Metal Nanoparticles: Synthesis, Characterization, and Applications to Catalysis” (Acc. Chem. Res. 2001, 34, 181–190) by R.M. Crooks et al.:

“An important distinction between PPI and PAMAM dendrimers is that the former are stable at very high temperatures (the onset of weight loss for G4 PPI is 470 °C), whereas PAMAM dendrimers undergo retro-Michael addition at temperatures higher than about 100 °C.”

Nonetheless, it should be noted, that PPI dendrimers undergo destruction (retro-Michael) under hydrogenation conditions at temperatures below 100 °C. Therefore, yes, it should be mentioned in the manuscript. The following sentences have been added:

“It should be noted, that PPI dendrimers may undergo retro-Michael addition under hydrogenation conditions even at temperatures below 100 °C [61]; as a consequence, dendrimer network carriers, at least, were partially decomposed at 350–400°C and 50 atm. of H2, under which the process was carried out. This might be additional reason, why in situ catalyst appeared more efficient.”

The paper is well-illustrated with many Schemes and Figures, but I have a few comments about some of them:

    In Figure 1, one can imagine that all dendrimers, except the phosphazene dendrimer, and the phenylazomethine dendrimer, are synthesized by divergent processes, because they have potentially reactive terminal functions. But reality is different. The poly(aryl ether) dendrimer is synthesized by a convergent way, and the phosphazene dendrimer by a divergent way. Thus, instead of presenting the poly(aryl ether) dendrimer, a poly(aryl ether) dendron should be presented, with phenyl groups (no OH) as terminal functions. On the other side, an aldehyde should be added in para position on all the terminal aryl groups of the phosphazene dendrimer.

Answer: This remark would be fair, if the present review was devoted to the synthesis and properties of dendrimers themselves. We do not discuss the questions above in the introduction and partially concern them only in Section 3.1, where the gradual dendron growth on the support is considered.

    In Figure 5, there are many overlaps between oxygen atoms, it is not very beautiful.

Answer: The Figure has been redrawn.

    In Schemes 9, 10, and 39, the drawings represent dendrons attached to the surface of nanoparticles, but the captions to these Schemes indicate that these are dendrimer-encapsulated nanoparticles. I suppose that the captions are right, and that the schemes should be modified. In the other Figures and Schemes, the difference between dendrons and dendrimer is clear, so it should be the same for these 3 Schemes.

Answer: The Schemes have been redrawn.

    In Figure 19, the real chemical structure of the polyaromatic framework is unclear. Why not drawing also the chemical structure?

Answer: The PAF structure in Figure 19 is similar to that in Angew. Chem. Int. Ed. 2009, 48, 9457–9460. Nonetheless, the image was changed.

    In Figure 21, the chemical structure of the dendrons is not given. It should be given.

Answer: Chemical structures of corresponding dendrons have been added to the Figure.

The paper is well organized, but the writing should be improved. There are many sentences which are exceedingly long (more than five lines, in some cases 10 lines), thus they should be split in at least two sentences. There are also a few systematic misprints:

    “Phenylazomethyne” should be replaced by “phenylazomethine” (6 occurrences)

    “Aminogroup” (or aminogroups) should be replaced by “amino group” (or amino groups) (12 occurrences)

    Delete “to” in “favoured to” (8 occurrences)

Answer: The said misprints are corrected.

And many other misprints:

    Lines 23-24: in the sentence “To prevent metal nanoparticle sintering and agglomeration, add, as a consequence, catalyst deactivation during the process, various organic ligands, such as ionic liquids”, “add” should be replaced by “and”

    Line 25: “thyols” should be replaced by “thiols”

    Line 167: replace “poly(ethyele imine)” by “poly(ethylene imine)”

    Line 273: replace “subnanoclasters” by “subnanoclusters”

Answer: Done

    Line 386: what is “Рукуштcarbon”?

Answer: This was “Herein carbon”. This misprint has been corrected.

    Line 571: replace “phospine“ by “phosphine”

    The sentence line 590 “SiO2-PAMAM-Rh catalysts appeared slight negative dendritic effect” should be modified such as “SiO2-PAMAM-Rh catalysts display a slightly negative dendritic effect”

    Lines 621-622: in “under to carbonylation”, delete “to”

    Line 625: replace “bromorenes“ by “bromoarenes“

    Line 630: replace “The same type catalysts” by “The same type of catalysts”

    Lines 633-634: “The maximum yields of 85–99% were observed for the substrates with electron withdrawing groups (C(=O)OMe, NO2), maintained at catalyst recycling” has to be modified, for instance as “The maximum yields of 85–99% were observed for the substrates with electron withdrawing groups (C(=O)OMe, NO2), and were maintained when recycling the catalyst”

    Line 745: in “conventional thermal impact“, replace “impact” by “heating”

    Line 771: replace “иby” by “by”

    Line 816: replace “the use of sterically hindrance ortho-substituted” by “the use of sterically hindered ortho-substituted”

    Line 883: replace “in differing ways” by “in different ways”

    Line 1054: replace “propogation” by “propagation”

    Line 1064: replace “80 0C” by “80°C”

    Line 1171: replace “ompregnation” by “impregnation”

    Lines 1307: in the sentence “Replace of 1-octyne”, “Replace” should become “Replacement”

    Line 1590: replace “PPI dendrimera” by “PPI dendrimer”

    Line 1622: replace “revealing” by “revealed”

    Line 1628: in “but resulted in the in the partial” delete one “in the”

    Line 1788: replace “dendonized” by “dendronized”

    Lines 1854-1855: replace “diphenylphoshite-terminated” by “diphenylphosphite-terminated”

    Line 1900: in “The third approach implies is the synthesis”, choose between “implies” or “is”, but don’t use both. And probably many other misprints I missed. This paper should be very carefully re-read by the authors.

Answer: Done. Several other misprints have been found at revision.

Reviewer 4 Report

This is a review article about dendrimer and metal catalysts.

Many papers are collected and each one are mentioned properly.

I am interested in this review.

However, some weak points should be fixed before acceptance.

1.  Classification are not clear.

For example,

   -Types of assembling ways of metal ions

   -Structure (blanch or connecting groups) of dendrimers

   -Organic reactions catalyzed by them

   -Purpose or application

were dealt with "randomly ordered" at a glance.

Please entitle appropriate "section names" and

arrange each section in a proper order with relevance. 

2. Reality or importance of drawings.

-Size of metal and organic moieties

-Representation of nanoclusters (a large circle or many small circles)

-Substrates of adsorption

Since these are drawn inconsistently (according to the original paper), I have the impression that it is difficult to understand what they represent and to understand their relationships.

3. The concept of "heterogeneous" is ambiguous or

impossible to catch up the its development.

Although entitled as a "heterogeneous" catalyst, the metal complexes introduced as a pendant group are typical homogeneous catalysts.

Also, if the metal nanoparticle catalyst is coordinated to the dendrimer

and coexists in the solution layer, it may work as a  "homogeneous"

catalyst. Please explain your opinion.

4. PAMAM and other dendrimers.

Certainly, PAMAM is a commercially available dendrimer that has been widely studied from the beginning and can ionically bond metals. Related to 1, but repeatedly covered in several sections. As a result, I get the impression that PAMAM and other compositions (which are rarely played) are unbalanced.

Moreover, all types of dendrimers in Fig.1 should be introduced from the 

viewpoints of each section.

That's all.

Author Response

Reviewer 4

This is a review article about dendrimer and metal catalysts.

Many papers are collected and each one are mentioned properly.

I am interested in this review.

Answer: Thank you for your kind revision.

However, some weak points should be fixed before acceptance.

  1. Classification are not clear. For example,

   -Types of assembling ways of metal ions

   -Structure (blanch or connecting groups) of dendrimers

   -Organic reactions catalyzed by them

   -Purpose or application

were dealt with "randomly ordered" at a glance. Please entitle appropriate "section names" and arrange each section in a proper order with relevance.

Answer: Indeed, dendrimer-based catalysts (as other dendritic materials) can be classified in different ways, e.g.: dendrimer type, application (reactions catalyzed) etc. Combination of these classifications can be imagined as a 3D matrix, whose transferring to the text inevitably results in some degree of disorder. Because this review is devoted to heterogeneous dendrimer-based catalysts, we have decided to order them by known heterogenization methods (adsorption/impregnation; chemical grafting to silica or carbon carriers; dendrimer cross-linking). Herein one may notice, that the Introduction consecutively considers dendrimer structure and common methods for the synthesis of dendrimer-based homogeneous catalysts (mono- and polymetallic dendrimers-encapsulated nanoparticles; nanoparticle-cored dendrimers) and several examples of their application are mentioned. Examples in the Sections 2 and 3.1 are considered in the following order: silica-supported catalysts, then carbon-supported catalysts. Section 3.2 begins with supported CNT-supported catalysts (on which Section 3.1 is finished) and then turns to silica carriers. Section 4 begins with coordination dendrimer networks and then considers covalently cross-linked dendrimer networks. Section 5 deals with magnetically separable catalysts and contains brief introduction and then the examples arranged in approximately in the following order: polyphenylene dendrimers à allyl-ended aryl-cored dendrimers à PAMAM dendrimers à poly(aryl ether) dendrimers à arylphosphite dendrimers. Herein the last example in the Section represents immobilization on the Co core with carbon coating, whereas previous examples deals with Fe3O4/SiO2 support.

This grouping by various carriers is additionally denoted by subsections in the manuscript. E.g. “2.1. Impregnation/adsorption on silica-based carriers”.

  1. Reality or importance of drawings.

-Size of metal and organic moieties

-Representation of nanoclusters (a large circle or many small circles)

-Substrates of adsorption

Since these are drawn inconsistently (according to the original paper), I have the impression that it is difficult to understand what they represent and to understand their relationships.

Answer: Proportions of metal nanoparticles and dendrimer moieties in the drawings are tentative. Else the drawings would be very large and intricate.

Nanoparticles are represented as many small circles, as a rule, when the corresponding part of the manuscript deals with bi-/polymetallic nanoparticles or at relatively large-scale figures and schemes, where the certain dendrimer type (e.g. PPI, PAMAM, PEI, DPA etc.) can be distinguished. Also in the drawings, where sc. nanoclusters (i.e. nanoperticles < 1 nm in diameter) are depicted. In other cases nanoparticles represented as large coloured circles, when dendrimers are just schematically presented, and nanoparticle size is unknown or does not matter.

  1. The concept of "heterogeneous" is ambiguous or impossible to catch up the its development. Although entitled as a "heterogeneous" catalyst, the metal complexes introduced as a pendant group are typical homogeneous catalysts. Also, if the metal nanoparticle catalyst is coordinated to the dendrimer and coexists in the solution layer, it may work as a "homogeneous" catalyst. Please explain your opinion.

Answer: We suppose, it depends on the certain dendrimer type, metal, linker etc. Dendrimer-stabilized and dendrimer-encapsulated nanoparticles can exist in solution and, therefore, act as a homogeneous catalyst. The similar statement can be referred to dendrimer metal complex catalysts, where the metal remaining coordination locations are occupied by the strong ligands, which cannot be displaced by dendrimer end group. Nonetheless, when dendrimers appear cross-linked by transition metal ions (coordination polymer, as in the ref. J. Mol. Catal. A: Chem. 2009, 297, 73–79) or organic bifunctional agents, they easily precipitate. Therefore, such catalyst will act as heterogeneous.

In whole, we assume, that heterogeneous dendritic catalysts exist and are recovered as solids, similar to conventional heterogeneous catalysts, but may act as pseudo-homogeneous in the reaction medium. We hope, it has become more understandable from this amplified paragraph of the Conclusion:

“Heterogenized dendrimer-based catalysts are obtained and stored as solids/powders similar to conventional heterogeneous catalysts. Nonetheless, various examples in hydrogenation, hydroformylation, oxidation, cross-coupling reactions etc. have revealed them not only to maintain their efficacy at recycling, being easily recoverable, but also to exhibit performance, often superior to analogous homogeneous catalysts, with TOF values reaching up to 400000 h–1. Herein dendrimer moieties acted as nanoreactors, preventing nanoparticle agglomeration and metal leaching and providing high reaction rate and selectivity. Reaction in these nanoreactors may proceed in a pseudo-homogeneous way on small nanoparticles and clusters (nanoheterogeneous catalysis) or on the metal complex centers, attached to heterogenized dendrimers or dendrons. “Release and catch mechanism”, suggesting the temporary process transition to homogeneous phase, can also take place. Moreover, polyamine dendrimers (PPI, PAMAM, PETIM) in water medium may act as acid-base catalysts, thus additionally increasing the reaction rate and directing the reaction pathway.”

  1. PAMAM and other dendrimers. Certainly, PAMAM is a commercially available dendrimer that has been widely studied from the beginning and can ionically bond metals. Related to 1, but repeatedly covered in several sections. As a result, I get the impression that PAMAM and other compositions (which are rarely played) are unbalanced.

Answer: We suppose, the reason is sufficiently simple. PAMAM dendrimers are not only commercially available, but also relatively simple synthesized, that is especially important at the gradual dendrimer growth on the support. Therefore PAMAM dendrimers are much frequently used for the catalyst design and, therefore, much frequently presented in the literature. More likely, we would prefer to discuss about amine dendrimers (PAMAM, PPI, PEI, PETIM etc.) in whole, because such dendrimers contain a lot of donor complex formation groups, that makes them more suitable for the design of both homogeneous and heterogeneous catalysts.

We have additionally noticed it in the Conclusion:

“Among others, polyamine dendrimers (PAMAM, PPI, PETIM, PEI) are characterized by commercial ability, relatively simple synthetic procedure and, first af all, the large amount of donor groups, being able to metal catching and complex formation; hence, the overwhelming majority of the catalyst examples, illustrated in the review, refers to dendrimers of the said types.”

Moreover, all types of dendrimers in Fig.1 should be introduced from the viewpoints of each section. That's all.

Answer: Why? Each example in the review is accompanied, as a rule, with the corresponding figure or scheme, where dendrimer of the certain type is presented. Moreover, Figure 1 in the Introduction illustrates the most known dendrimer types. Introduction explains, what is dendrimer, how it is structured, how homogeneous dendrimer-based catalysts are synthesized and in what processes can be applied. However, not all dendrimer types are suitable for the design of heterogeneous catalysts and, therefore, cannot be illustrated elsewhere in the review, except of Introduction.